

# On 2d CFTs that interpolate between minimal models

**Sylvain Ribault**

Institut de physique théorique, CNRS, CEA, Université Paris-Saclay, France

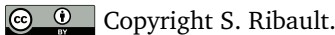

sylvain.ribault@ipht.fr

## Abstract

We investigate exactly solvable two-dimensional conformal field theories that exist at generic values of the central charge, and that interpolate between A-series or D-series minimal models. When the central charge becomes rational, correlation functions of these CFTs may tend to correlation functions of minimal models, or diverge, or have finite limits which can be logarithmic. These results are based on analytic relations between four-point structure constants and residues of conformal blocks.

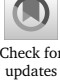
# 1 Introduction and summary

## 1.1 Motivations

Thanks to their infinite-dimensional symmetry algebras, two-dimensional conformal field theories can in some cases be classified and solved. This not only benefits their own applications, but also provides lessons for the study of higher-dimensional conformal field theories, for which exact results are much harder to derive.

The simplest nontrivial two-dimensional CFTs are the Virasoro minimal models: rational CFTs that exist at discrete values of the central charge, and can be either diagonal (A-series) or not (D-series and E-series). Other solvable CFTs of comparable complexity are known to exist at arbitrary complex central charges, namely Liouville theory and generalized minimal models. Both Liouville theory and generalized minimal models are diagonal, i.e. their spectrums are of the type $\oplus_i \mathcal{R}_i \otimes \bar{\mathcal{R}}_i$, where each term involves the same irreducible representation for the left-moving Virasoro algebra as for the right-moving Virasoro algebra. Until recently, it was not clear whether solvable, non-diagonal CFTs could be constructed at generic central charges.

Then, when trying to describe cluster connectivities in the Potts model (a model which exists at least for central charges $c \in (-2, 1)$), we stumbled upon a crossing-symmetric four-point function whose spectrum was non-diagonal and could be determined exactly [1]. In subsequent work, we have found large classes of four-point functions with the same spectrum, and exactly determined the structure constants [2]. These four-point functions actually exist for any central charge such that $\Re c < 13$. For $c \in (-\infty, 1)$, we have argued that they belong to CFTs that can be constructed as limits of D-series minimal models.

Conversely, in the present work, we will show that (under certain conditions) the new non-diagonal CFTs reduce to D-series minimal models when the central charge becomes rational. This is interesting for the following reasons:

1. Some features of these CFTs, such as OPEs between two non-diagonal fields, are still poorly understood: the reduction to minimal models elucidates such features at rational central charges.

2. These CFTs then provide approximations of minimal models, which resolve the singularities that plague computations at rational central charges.

3. We will obtain a unified picture of D-series minimal models as special cases of CFTs that depend smoothly on the central charge. Having a picture of the space of consistent CFTs, and not just of isolated points such as minimal models, is particularly important when using solvable CFTs as testing grounds for numerical bootstrap techniques [3].

The last two motivations apply not only to non-diagonal CFTs, but to diagonal CFTs as well, and we will investigate to what extent generalized minimal models reduce to A-series minimal models when the central charge becomes rational.

## 1.2 The models under consideration

Let us introduce the CFTs that we will consider, by writing their spectrums. The spectrum of a two-dimensional CFT is a representation of the product of the left-moving and right-moving Virasoro algebras. Both Virasoro algebras have the same central charge $c$, which we will write in terms of the number $\beta^2$ such that

$$c = 1 - 6\left(\beta - \frac{1}{\beta}\right)^2 \quad \text{with} \quad |\beta| \leq 1 \,. \tag{1}$$

We will write spectrums as combinations of irreducible highest-weight representations of the Virasoro algebra. Two types of representations will appear:

- Verma modules $\mathcal{V}_P$, with momentums $P$ related to conformal dimensions $\Delta(P)$ by

$$\Delta(P) = \frac{c-1}{24} + P^2 \,. \tag{2}$$

- Degenerate representations $\mathcal{R}_{\langle r,s \rangle}$ with $r, s \in \mathbb{N}^*$, with dimensions and momentums of the type

$$\Delta_{\langle r,s \rangle} = \Delta(P_{\langle r,s \rangle}) \quad \text{with} \quad P_{\langle r,s \rangle} = \frac{1}{2}\left(\beta r - \frac{s}{\beta}\right) \,. \tag{3}$$

We will investigate the relations between CFTs that exist for $\beta^2$ irrational, and minimal models, which exist for

$$\beta^2 = \frac{p}{q} \qquad \text{with} \qquad 2 \leq p < q \ \text{coprime integers} \,. \tag{4}$$

The spectrum of a minimal model is built from the degenerate representations that appear in its Kac table. The Kac table is usually written as the finite set of integer indices $(r,s) \in [1, p-1] \times [1, q-1]$. Taking advantage of the identity of conformal dimensions

$$\forall \lambda \in \mathbb{C} \,, \quad \Delta_{\langle r,s \rangle} = \Delta_{\langle r+\lambda q, s+\lambda p \rangle} \,, \tag{5}$$

we will formally write the identities of representations $\mathcal{R}_{\langle r,s \rangle} = \mathcal{R}_{\langle r-\frac{q}{2}, s-\frac{p}{2} \rangle}$, and rewrite the Kac table as a set of half-integer indices, centered on $(0,0)$,

$$K_{p,q} = \left[\left(\mathbb{Z} + \tfrac{q}{2}\right) \cap \left(-\tfrac{q}{2}, \tfrac{q}{2}\right)\right] \times \left[\left(\mathbb{Z} + \tfrac{p}{2}\right) \cap \left(-\tfrac{p}{2}, \tfrac{p}{2}\right)\right] \,. \tag{6}$$

With these notations, the spectrums of the A-series (diagonal) and D-series (non-diagonal) minimal models are

$$\mathcal{S}_{p,q}^{\text{A-series}} = \frac{1}{2} \bigoplus_{(r,s) \in K_{p,q}} \left|\mathcal{R}_{\langle r,s \rangle}\right|^2 \,, \tag{7}$$

$$\mathcal{S}_{p,q}^{\text{D-series}} = \frac{1}{2} \bigoplus_{\substack{(r,s) \in K_{p,q} \\ rs \in \mathbb{Z} + \frac{1}{2} + \frac{pq}{4}}} \left|\mathcal{R}_{\langle r,s \rangle}\right|^2 \oplus \frac{1}{2} \bigoplus_{\substack{(r,s) \in K_{p,q} \\ rs \in \mathbb{Z}}} \mathcal{R}_{\langle r,s \rangle} \otimes \bar{\mathcal{R}}_{\langle -r,s \rangle} \,, \tag{8}$$

where the factors $\frac{1}{2}$ eliminate the redundancy that comes from $\Delta_{\langle r,s \rangle} = \Delta_{\langle -r,-s \rangle}$. (The D-series model actually reduces to the A-series model if $p, q$ are both odd, and also if one of them is 2 or 4.)

The CFTs that exist for generic $\beta^2$ and that we will relate to minimal models are called the generalized minimal models (diagonal), and the odd and even CFTs (non-diagonal). Their spectrums are

$$\mathcal{S}^{\text{GMM}}_{\beta^2} = \frac{1}{2} \bigoplus_{(r,s)\in\mathbb{N}^*} \left| \mathcal{R}_{\langle r,s\rangle}\right|^2 \underset{\substack{\beta^2>0}}{=} \lim_{\substack{\frac{p}{q}\to\beta^2 \\ \text{fixed indices in } \mathbb{N}^*}} \mathcal{S}^{\text{A-series}}_{p,q} \,, \tag{9}$$

$$\mathcal{S}^{\text{odd}}_{\beta^2} = \mathcal{S}^{\text{Liouville}} \oplus \frac{1}{2} \bigoplus_{r\in2\mathbb{Z}} \bigoplus_{s\in\mathbb{Z}+\frac{1}{2}} \mathcal{V}_{P_{\langle r,s\rangle}} \otimes \bar{\mathcal{V}}_{P_{\langle -r,s\rangle}} \underset{\substack{\beta^2>0}}{=} \lim_{\substack{\frac{p}{q}\to\beta^2 \\ p \text{ odd}}} \mathcal{S}^{\text{D-series}}_{p,q} \,, \tag{10}$$

$$\mathcal{S}^{\text{even}}_{\beta^2} = \mathcal{S}^{\text{Liouville}} \oplus \frac{1}{2} \bigoplus_{r\in\mathbb{Z}+\frac{1}{2}} \bigoplus_{s\in2\mathbb{Z}} \mathcal{V}_{P_{\langle r,s\rangle}} \otimes \bar{\mathcal{V}}_{P_{\langle -r,s\rangle}} \underset{\substack{\beta^2>0}}{=} \lim_{\substack{\frac{p}{q}\to\beta^2 \\ p \text{ even}}} \mathcal{S}^{\text{D-series}}_{p,q} \,, \tag{11}$$

where $\mathcal{S}^{\text{Liouville}} = \int_{\mathbb{R}_+} dP \, \mathcal{V}_P \otimes \bar{\mathcal{V}}_P$ is the diagonal, continuous spectrum of Liouville theory. Generalized minimal models actually exist for $\beta^2 \in \mathbb{C} - \mathbb{Q}$, while the even and odd CFTs exist for $\beta^2 \in (\mathbb{C} - \mathbb{Q}) \cap \{\Re\beta^2 > 0\}$.

In order to analyze limits of CFTs, it is not enough to consider spectrums: we should also study four-point correlation functions. In the spirit of the conformal bootstrap approach, four-point functions indeed encode all relevant information on a CFT on the sphere, and in principle allow us to reconstruct all other correlation functions. We will compute a four-point function $\langle V_1 V_2 V_3 V_4 \rangle$ using its $s$-channel decomposition into structure constants and conformal blocks,

$$\langle V_1 V_2 V_3 V_4 \rangle = \sum_{s\in\mathcal{S}_{1234}} D_s \mathcal{F}_{\Delta_s} \bar{\mathcal{F}}_{\bar{\Delta}_s} \,, \tag{12}$$

where $\mathcal{F}_{\Delta_s}$ and $\bar{\mathcal{F}}_{\bar{\Delta}_s}$ are left- and right-moving $s$-channel conformal blocks respectively, and $D_s$ are the four-point structure constants. The index $s$ runs over a subset $\mathcal{S}_{1234}$ of the spectrum. This subset can be discrete or continuous, depending on the operator product expansion $V_1 V_2$, equivalently on the fusion rules of the corresponding representations.

In Section 2 we will give a more complete review of our CFTs and their correlation functions, in particular for $\beta^2 \in \mathbb{R}_{>0}$ we will construct the even and odd CFTs and the generalized minimal models as limits of minimal models.

## 1.3 The results

We will study the behaviour of four-point functions (12) in the even and odd CFTs and the generalized minimal models in the limits $\beta^2 \to \frac{p}{q}$. We will begin with separately analyzing the behaviour of conformal blocks and structure constants, before bringing them together. When bringing them together, we will observe many nontrivial simplifications, whose technical basis lies in expressions for both the structure constants (29) and residues of conformal blocks (40) in terms of the same special functions. Our results will be mostly conjectures, because we only analyze the first few terms of infinite $s$-channel decompositions, and of Zamolodchikov's expression for conformal blocks as infinite series: this is enough for guessing the behaviour at all orders, but it remains to systematically understand the combinatorics of these simplifications.

Let us summarize the main results:

- In Conjecture 3.1, we describe how minimal model conformal blocks are obtained as limits of conformal blocks with generic conformal dimensions and/or central charge.

- In Proposition 4.1, we characterize the zeros of three-point structure constants of the odd and even CFTs, as functions of the central charge.

- We bring these results together in Conjecture 4.2, which states that certain four-point functions in the odd and even CFTs only have simple poles as functions of the central charge, although they are infinite sums of terms that can have poles of unbounded orders.

- From this technical result, we then deduce that any four-point function with two diagonal and two non-diagonal fields in a D-series minimal model, is a $\beta^2 \to \frac{p}{q}$ limit of four-point functions in the odd or even CFT, depending on the parity of $p$. (Conjecture 4.3.)

- We finally focus on diagonal CFTs, and study the limits $\beta^2 \to \frac{p}{q}$ of four-point functions in generalized minimal models. In contrast to the non-diagonal case, we find that we do not recover the A-series minimal model whenever all four fields belong to its Kac table $K_{p,q}$: Conjecture 5.1 only states that the limit is finite. In some cases the limit is a four-point function in the minimal model, in other cases it may well belong to some other CFT, possibly logarithmic and/or non-diagonal.

This means that generalized minimal models interpolate between A-series minimal models, and that the even and odd CFTs interpolate between D-series minimal models: not in all cases, but for certain choices of correlation functions and of minimal models.

## 1.4 Outlook

On the practical side, our results imply that we can approximate correlation functions in minimal models by slightly perturbing the central charge. Conformal blocks and structure constants can have singularities at rational central charges: then the perturbation removes these singularities, and acts as a regulator. Our results also suggest that we should impose minimal model fusion rules by hand, rather than wait for them to emerge in the rational limit: not only because they do not always emerge in the A-series case, but also because their emergence can depend on cancellations between finite or even divergent terms.

It would be interesting to investigate more general four-point functions in rational central charge limits. To begin with, if we wanted to understand how a given four-point function behaves at all rational central charges, we would have to study what happens in limits where at least some of the fields are outside the Kac table. Moreover, it would be interesting to study the limits of four-point functions with 0 or 4 non-diagonal fields, in addition to the four-point functions with 2 diagonal fields. However, we would first need to determine the operator product expansion of two non-diagonal fields in the odd and even CFT: a difficult problem in its own right.

Our broader message is that CFTs that exist at rational central charges, can often be derived from CFTs that exist at generic central charges. This is a priori interesting, because at rational central charges Virasoro representations have complicated structures, and conformal blocks have singularities: these problems are milder or absent at generic central charges. We do not necessarily expect that all CFTs at rational central charges can be derived in this manner, and in particular we do not know how to derive E-series minimal models. But in contrast to the other series, E-series minimal models have central charges that are not dense in $(-\infty, 1)$: in this sense, we can derive almost all minimal models. (Lest we are accused of circular reasoning, we insist that the even and odd CFTs can be constructed independently of D-series minimal models [2].)

## 2 Solvable CFTs as limits of minimal models

In this Section, we review the construction of the odd and even CFTs as limits of D-series minimal models [2]. In particular, we write the exact expressions for the structure constants of these CFTs.

### 2.1 Minimal models

We start with a review of the minimal models themselves. Any two coprime integers such that $2 \leq p < q$ label an A-series minimal model. If moreover one of the integers belongs to $3 + 2\mathbb{N}$, and the other one belongs to $6 + 2\mathbb{N}$, then they also label a D-series minimal model.

We have already written the spectrums of minimal models in Eqs. (7) and (8). In order to characterize correlation functions, let us sketch the fusion rules and operator product expansions of these models. We will again use notations such that the Kac table is a rectangle whose center is the origin. In these notations, the fusion rules of the degenerate representations $\mathcal{R}_{\langle r,s \rangle}$ of the Virasoro algebra that appear in the Kac table are,

$$\mathcal{R}_{\langle r_1,s_1 \rangle} \times \mathcal{R}_{\langle r_2,s_2 \rangle} = \bigoplus_{r \overset{2}{=} 1 - \frac{q}{2} + |r_1 - r_2|}^{\frac{q}{2} - 1 - |r_1 + r_2|} \bigoplus_{s \overset{2}{=} 1 - \frac{p}{2} + |s_1 - s_2|}^{\frac{p}{2} - 1 - |s_1 + s_2|} \mathcal{R}_{\langle r,s \rangle} , \tag{13}$$

where the notation $\overset{2}{=}$ is for sums that run by increments of 2. Equivalently, the condition that three Kac table representations $\mathcal{R}_{\langle r_i,s_i \rangle}$ are intertwined by fusion can be written in a manifestly permutation-invariant form,

$$\exists \epsilon \in \{\pm 1\} | \forall (\epsilon_1, \epsilon_2, \epsilon_3) \in \{\pm 1\}^3 , \ \epsilon_1 \epsilon_2 \epsilon_3 = \epsilon \implies \begin{cases} \frac{q}{2} + \sum_i \epsilon_i r_i & \in 2\mathbb{N} + 1 , \\ \frac{p}{2} + \sum_i \epsilon_i s_i & \in 2\mathbb{N} + 1 . \end{cases} \tag{14}$$

This condition on three pairs of indices $(r_i, s_i)$ actually implies that all of them belong to the Kac table.

While fusion rules are statements about representations of the Virasoro algebra, states and fields of our models belong to representations of the product of a left- and right-moving Virasoro algebras. In order to describe operator product expansions, we must therefore supplement fusion rules with information on how left- and right-moving representations interact. Calling $V_{\langle r,s \rangle}^D$ and $V_{\langle r,s \rangle}^N$ the diagonal and non-diagonal fields of our D-series minimal models, their OPEs are determined by the requirements that fusion rules are respected, and diagonality is conserved. For example, the OPE of a diagonal field with a non-diagonal field is a combination of non-diagonal fields,

$$V_{\langle r_1,s_1 \rangle}^D V_{\langle r_2,s_2 \rangle}^N \sim \sum_{r \overset{2}{=} 1 - \frac{q}{2} + |r_1 - r_2|}^{\frac{q}{2} - 1 - |r_1 + r_2|} \sum_{s \overset{2}{=} 1 - \frac{p}{2} + |s_1 - s_2|}^{\frac{p}{2} - 1 - |s_1 + s_2|} V_{\langle r,s \rangle}^N , \tag{15}$$

where the notation $\overset{2}{=}$ is for sums that run by increments of 2. Having written this $V^D V^N \sim V^N$ OPE, we trust that we need not explicitly write the $V^D V^D \sim V^D$ and $V^N V^N \sim V^D$ OPEs.

### 2.2 Non-rational limits

When the integer parameters $p, q$ of D-series minimal models vary, the parameter $\beta^2 = \frac{p}{q}$ (4) takes values that are dense in $(0, 1)$. Each value of $\beta_0^2 \in (0, 1)$ can be approached by fractions with either $p$ odd, or $p$ even, giving rise to the odd and even limits of D-series minimal models.

It was conjectured that both limits exist [2]. Therefore, for any $\beta_0^2 \in (0,1)$, there exist two distinct limiting CFTs, which we call the **odd and even CFTs**.

Let us review how the spectrum behaves in these limits. The fundamental feature of the degenerate representation $\mathcal{R}_{\langle r,s \rangle}$ with $r, s \in \mathbb{N}^*$ is that it has a vanishing null vector at the level $rs$, with therefore the conformal dimension

$$\Delta_{\langle r,-s \rangle} = \Delta_{\langle r,s \rangle} + rs \ . \tag{16}$$

Actually, if $\beta^2 = \frac{p}{q}$, we have $\mathcal{R}_{\langle r,s \rangle} = \mathcal{R}_{\langle q-r,p-s \rangle}$ due to eq. (5), and therefore a second vanishing null vector at the level $(q-r)(p-s)$. In the limit $p, q \to \infty$ with $r, s$ fixed, the second null vector disappears, and we are left with a degenerate representation with only one vanishing null vector:

$$\lim_{\substack{\frac{p}{q} \to \beta_0^2 \\ r,s \in \mathbb{N}^* \text{ fixed}}} \mathcal{R}_{\langle r,s \rangle} = \mathcal{R}_{\langle r,s \rangle} \ . \tag{17}$$

Let us apply this limit to the spectrums (7) of the A-series minimal models. Since $\lim_{p,q\to\infty}[1, q-1] \times [1, p-1] = \mathbb{N}^* \times \mathbb{N}^*$, we obtain the spectrum (9) of the generalized minimal model. The fusion rules also simplify in this limit, and we recover the fusion rules of degenerate representations at generic central charges,

$$\mathcal{R}_{\langle r_1,s_1 \rangle} \times \mathcal{R}_{\langle r_2,s_2 \rangle} = \bigoplus_{r \overset{2}{=} |r_1 - r_2|+1}^{r_1+r_2-1} \bigoplus_{s \overset{2}{=} |s_1 - s_2|+1}^{s_1+s_2-1} \mathcal{R}_{\langle r,s \rangle} \ . \tag{18}$$

This suggests that the limits of A-series minimal models are generalized minimal models.

When it comes to D-series minimal models, we cannot take a limit where the integer indices of degenerate representations would be fixed. This is because the non-diagonal sector of the spectrum (8) is a sum of representations of the type $\mathcal{R}_{\langle r,s \rangle} \otimes \bar{\mathcal{R}}_{\langle q-r,s \rangle}$, with $(r,s) \in [1, p-1] \times [1, q-1]$. This is actually the reason why we wrote the spectrum as combinations of representations of the type $\mathcal{R}_{\langle r,s \rangle} \otimes \bar{\mathcal{R}}_{\langle -r,s \rangle}$, at the cost of allowing non-integer indices. In this notation, the representation $\mathcal{R}_{\langle r,s \rangle}$ has vanishing null vectors at the levels $(\frac{q}{2} + r)(\frac{p}{2} + s)$ and $(\frac{q}{2} - r)(\frac{p}{2} - s)$. These levels go to infinity if we keep $(r,s) \in K_{p,q}$ fixed while $p, q \to \infty$, where the Kac table $K_{p,q}$ was given in eq. (6). Then our representation tends to the Verma module with the same momentum,

$$\lim_{\substack{\frac{p}{q} \to \beta_0^2 \\ (r,s) \in K_{p,q} \text{ fixed}}} \mathcal{R}_{\langle r,s \rangle} = \mathcal{V}_{P_{\langle r,s \rangle}} \ . \tag{19}$$

It is now straightforward to compute the limit of the non-diagonal sector of the spectrum. The only subtlety is that we have to choose which one of the two minimal model indices $p, q$ is odd, and which one is even. Depending on this choice, we obtain two different limits: the non-diagonal sectors of the odd (10) and even (11) CFTs. For example, if $p$ is odd, then an element of the Kac table $(r,s) \in K_{p,q}$ has a half-integer first index $r \in \mathbb{Z} + \frac{1}{2}$. The condition $rs \in \mathbb{Z}$ from the non-diagonal sector of the spectrum (8) then implies $s \in 2\mathbb{Z}$.

Taking our limit is more subtle in the diagonal sector than in the non-diagonal sector, because the diagonal representation $\mathcal{R}_{\langle r,s \rangle} \otimes \bar{\mathcal{R}}_{\langle r,s \rangle}$ depends on $r, s$ solely through the combination $P_{\langle r,s \rangle}$. We will therefore study the distribution of the momentums $P_{\langle r,s \rangle}$, as was first done in the case $\beta_0^2 = 1$ by Runkel and Watts [4]. In both our even and odd limits, the momentums $P_{\langle r,s \rangle}$ become uniformly distributed on the real line, and we have

$$\lim_{\frac{p}{q} \to \beta_0^2} \mathcal{S}_{p,q}^{\text{D-series, diagonal}} \propto \int_{\mathbb{R}_+} dP \ \mathcal{V}_P \otimes \bar{\mathcal{V}}_P \ . \tag{20}$$

The unknown proportionality coefficient is the multiplicity of representations in the limit diagonal spectrum: this should be an integer, possibly infinite. It will turn out that in the limit theory, correlation functions depend solely on the fields' conformal dimensions, and this means that the multiplicity is one. Therefore, the limit diagonal spectrum coincides with the spectrum of Liouville theory [5], and the limits (10), (11) hold for the full spectrum, not just the non-diagonal sector.

Let us discuss how the OPEs behave in our limits. When $p, q \to \infty$, the fusion rules (13) simply lose their bounds on the summed indices. As a result, the only constraint on OPEs that survives is the conservation of diagonality. In the odd theory, the limiting OPEs are therefore

$$V_{P_1}^D V_{P_2}^D \sim \int_{\mathbb{R}_+} dP \, V_P^D \,, \tag{21}$$

$$V_{P_1}^D V_{\langle r_2, s_2 \rangle}^N \sim \sum_{r \in 2\mathbb{Z}} \sum_{s \in \mathbb{Z} + \frac{1}{2}} V_{\langle r, s \rangle}^N \,, \tag{22}$$

$$V_{\langle r_1, s_1 \rangle}^N V_{\langle r_2, s_2 \rangle}^N \sim \int_{\mathbb{R}_+} dP \, V_P^D \,. \tag{23}$$

However, these formal limits of OPEs do not necessarily coincide with the OPEs of the even and odd CFTs, because taking our limits does need not necessarily commute with taking OPEs. A finer analysis of the behaviour of correlation functions would be needed in order to reliably derive the OPEs of the even and odd CFTs, and it is not even clear that diagonality is actually conserved. Nevertheless, we know that the $V^D V^N$ OPE (22) is correct, because it leads to crossing-symmetric four-point functions of the type $\langle V^D V^N V^D V^N \rangle$ [2]. It is such four-point functions that we will analyze in Section 4.

## 2.3 Structure constants

In any CFT, arbitrary correlation functions can be reduced to combinations of conformal blocks, and two- and three-point functions. (This is a consequence of the existence of OPEs.) In minimal models and in our odd and even CFTs, two- and three-point functions are known explicitly [2], and we will now review them. We adopt a particular field normalization, namely the normalization such that $Y = 1$ in the notations of [2]. Our results do not depend on this choice. Moreover, since two- and three-point functions have universal dependences on field positions, we will keep these dependences implicit, and identify two- and three-point functions with the corresponding structure constants.

For a two-point function to be nonzero, the left and right conformal dimensions of the two fields must be the same. We are dealing with CFTs where the Virasoro generator $L_0$ is diagonalizable, so there is a basis of fields whose two-point functions are of the type

$$\langle V_1 V_2 \rangle = \delta_{12} \langle V_1 V_1 \rangle \,. \tag{24}$$

Here $\langle V_1 V_1 \rangle$ is a function of the left and right momentums of the field, namely

$$\langle VV \rangle = \frac{(-1)^{P^2 - \bar{P}^2}}{\prod_{\pm} \Gamma_\beta(\beta \pm 2P) \Gamma_\beta(\beta^{-1} \pm 2\bar{P})} \,. \tag{25}$$

This expression uses the double Gamma function $\Gamma_\beta$. We refrain from defining this function or giving its basic properties, since this information is available in Wikipedia.

Unlike the two-point function, the three-point function does not depend solely on the momentums of the fields, but also on whether they are diagonal or non-diagonal. We will now write the three-point function for one diagonal and two non-diagonal fields: this will be

enough for computing four-point functions of the type $\langle V^D V^N V^D V^N \rangle$. We assume that both non-diagonal fields belong to a minimal model, to the odd CFT, or to the even CFT, i.e. that they have indices $(r,s)$ in $2\mathbb{Z} \times (\mathbb{Z} + \frac{1}{2})$ or $(\mathbb{Z} + \frac{1}{2}) \times 2\mathbb{Z}$. On the other hand, the diagonal fields can have arbitrary momenta. The three-point function is then

$$\left\langle V_1^D V_2^N V_3^N \right\rangle = \frac{1}{\prod_{\pm,\pm} \Gamma_\beta(\frac{\beta}{2} + \frac{1}{2\beta} + P_1 \pm P_2 \pm P_3) \prod_{\pm,\pm} \Gamma_\beta(\frac{\beta}{2} + \frac{1}{2\beta} - P_1 \pm \bar{P}_2 \pm \bar{P}_3)} \ . \tag{26}$$

(The original formula in ref. [2] has an extra sign factor; in our case this factor only depends on the diagonal field, and can be absorbed in its normalization.) In the case of three diagonal fields, the three-point function is still given by eq. (26). It can we written more compactly using the function $\Upsilon_\beta(x) = \frac{1}{\Gamma_\beta(x)\Gamma_\beta(\beta+\beta^{-1}-x)}$, namely

$$\left\langle V_1^D V_2^D V_3^D \right\rangle = \prod_{\pm,\pm} \Upsilon_\beta \left( \frac{\beta}{2} + \frac{1}{2\beta} + P_1 \pm P_2 \pm P_3 \right) \ . \tag{27}$$

From the two- and three-point functions, we can build the four-point structure constants, i.e. the coefficients $D_s$ of the four-point function's decomposition into conformal blocks (12),

$$D_s = \frac{\langle V_1 V_2 V_s \rangle \langle V_3 V_4 V_s \rangle}{\langle V_s V_s \rangle} \ . \tag{28}$$

We now assume that our four-point function is of the type $\left\langle V_1^D V_2^N V_3^D V_4^N \right\rangle$, so that our $s$-channel fields are non-diagonal and belong to a discrete set. Let us introduce a factorization of the four-point structure constants into left- and right-moving factors,

$$D_s = d_+(P_s) \bar{d}_-(\bar{P}_s) \ . \tag{29}$$

This factorization will be important in the following, because we will express conformal blocks in terms of the same functions $d_\pm$. We define these functions as

$$d_+(P_s) = \frac{e^{i\pi P_s^2} \prod_{\pm} \Gamma_\beta(\beta \pm 2P_s)}{\prod_{\pm,\pm} \Gamma_\beta(\frac{\beta}{2} + \frac{1}{2\beta} + P_1 \pm P_2 \pm P_s) \prod_{\pm,\pm} \Gamma_\beta(\frac{\beta}{2} + \frac{1}{2\beta} + P_3 \pm P_4 \pm P_s)} \ , \tag{30}$$

$$\bar{d}_-(\bar{P}_s) = \frac{e^{-i\pi \bar{P}_s^2} \prod_{\pm} \Gamma_\beta(\beta^{-1} \pm 2\bar{P}_s)}{\prod_{\pm,\pm} \Gamma_\beta(\frac{\beta}{2} + \frac{1}{2\beta} - P_1 \pm \bar{P}_2 \pm \bar{P}_s) \prod_{\pm,\pm} \Gamma_\beta(\frac{\beta}{2} + \frac{1}{2\beta} - P_3 \pm \bar{P}_4 \pm \bar{P}_s)} \ , \tag{31}$$

where the bar over $\bar{d}_-$ indicates that we should use the right-moving momentums $\bar{P}_2, \bar{P}_4$.

## 2.4 Analytic continuation

Although we are mainly concerned with rational limits, and therefore with central charges in the line $c \in (-\infty, 1)$, let us discuss the analytic continuation of the even and odd CFTs to complex central charges, if only to complete the picture. In the $s$-channel decomposition (12) of four-point function is of the type $\left\langle V_1^D V_2^N V_3^D V_4^N \right\rangle$, the structure constants and conformal blocks depend analytically on $\beta$, which makes the analytic continuation possible. However, the sum converges only if the real part of the total conformal dimension is bounded from below. The total conformal dimension of a non-diagonal field $V_{\langle r,s \rangle}^N$ is

$$\Delta_{\langle r,s \rangle} + \Delta_{\langle r,-s \rangle} = \frac{c-1}{12} + \frac{1}{2} \left( \beta^2 r^2 + \beta^{-2} s^2 \right) \ . \tag{32}$$

For $(r,s) \in 2\mathbb{Z} \times (\mathbb{Z} + \frac{1}{2})$, this is bounded from below provided $\Re\beta^2 > 0$ i.e. $\Re c < 13$.

Now we have defined the parameter $\beta$ such that $|\beta| < 1$ (1), but what happens if we analytically continue through the circle $|\beta| = 1$? Nothing dramatic, since our correlation functions are smooth functions of $\beta$. But according to our terminology, the odd CFT turns into the even CFT and vice-versa. Therefore, we can view both CFTs as two cases of a unique CFT that lives on the half-plane $\{\Re\beta^2 > 0\}$, equivalently on the double cover of the half-plane $\{\Re c < 13\}$. We then have a change of terminology across $|\beta| = 1$, equivalently across the branch cut $c \in (1, 13)$.

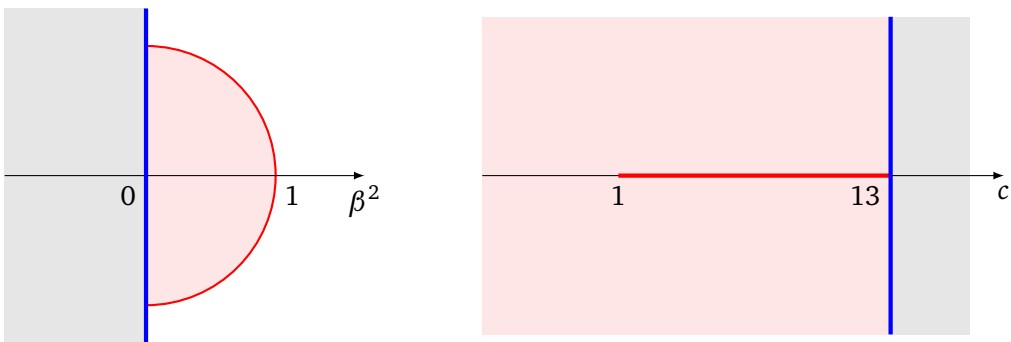

Figure 1: The complex $\beta^2$- and $c$-planes, with the boundaries of the domains of definition of the even and odd CFTs in blue, and the name-changing lines in red.

## 3 Rational limits of conformal blocks

### 3.1 Recursive representation

The $s$-channel conformal block $\mathcal{F}_\Delta$ is a function not only of the conformal dimension $\Delta$, but also of the central charge $c$ and of the dimensions $\{\Delta_i\}$ and positions $\{z_i\}$ of the four fields $V_1, \ldots, V_4$. (See [6] for a review.) Let us write it as

$$\mathcal{F}_\Delta(\{z_i\}) = \mathcal{F}^{(0)}(\{z_i\})\rho^\Delta H_\Delta(\rho) . \tag{33}$$

Here $\mathcal{F}^{(0)}(\{z_i\})$ is a $\Delta$-independent prefactor that depends analytically on $c$ and $\{\Delta_i\}$, $\rho$ is a function of $\{z_i\}$ (namely 16 times the elliptic nome), and the function $H_\Delta(\rho)$ is determined by Zamolodchikov's recursive representation

$$H_\Delta(\rho) = 1 + \sum_{m,n=1}^{\infty} \frac{\rho^{mn}}{\Delta - \Delta_{\langle m,n \rangle}} R_{m,n} H_{\Delta_{\langle m,-n \rangle}}(\rho) . \tag{34}$$

This is an entire function of the type

$$H_\Delta(\rho) = 1 + \sum_{N=1}^{\infty} H_\Delta^N \rho^N , \tag{35}$$

where the integer $N$ is called the level. The recursive representation shows that as a function of $\Delta$, the conformal block has simple poles at the degenerate dimensions $\{\Delta_{\langle m,n \rangle}\}$. The residues of these poles involve coefficients $R_{m,n}$ that we will call residues themselves, and that are given by the formula

$$R_{m,n} = \frac{-2P_{\langle 0,0 \rangle}P_{\langle m,n \rangle}}{\prod_{r=1-m}^{m} \prod_{s=1-n}^{n} 2P_{\langle r,s \rangle}} \prod_{r \stackrel{2}{=} 1-m}^{m-1} \prod_{s \stackrel{2}{=} 1-n}^{n-1} \prod_{\pm} (P_2 \pm P_1 + P_{\langle r,s \rangle})(P_3 \pm P_4 + P_{\langle r,s \rangle}) , \tag{36}$$

where the degenerate momentums $P_{\langle r,s\rangle}$ are defined in eq. (3). Let us write these residues in terms of the Barnes double Gamma function $\Gamma_\beta(x)$. From $x = \frac{\Gamma(x+1)}{\Gamma(x)}$ and $\Gamma(\beta x) = \sqrt{2\pi}\beta^{\beta x - \frac{1}{2}}\frac{\Gamma_\beta(x)}{\Gamma_\beta(x+\beta)}$, we deduce the identities

$$\prod_{r\overset{2}{=}1-m}^{m-1}\prod_{s\overset{2}{=}1-n}^{n-1}(x+P_{\langle r,s\rangle}) = \frac{\Gamma_\beta(\frac{\beta}{2}+\frac{1}{2\beta}+x+P_{\langle m,n\rangle})\Gamma_\beta(\frac{\beta}{2}+\frac{1}{2\beta}+x+P_{\langle -m,-n\rangle})}{\Gamma_\beta(\frac{\beta}{2}+\frac{1}{2\beta}+x+P_{\langle -m,n\rangle})\Gamma_\beta(\frac{\beta}{2}+\frac{1}{2\beta}+x+P_{\langle m,-n\rangle})} , \tag{37}$$

as well as

$$\frac{1}{2P_{\langle 0,0\rangle}}\prod_{r=1-m}^{m}\prod_{s=1-n}^{n}2P_{\langle r,s\rangle} = \frac{\Gamma_\beta(\beta+2P_{\langle m,n\rangle})\Gamma_\beta(\beta+2P_{\langle -m,-n\rangle})}{\operatorname{Res}\Gamma_\beta(\beta+2P_{\langle -m,n\rangle})\Gamma_\beta(\beta+2P_{\langle m,-n\rangle})} , \tag{38}$$

$$= -\frac{\Gamma_\beta(\beta^{-1}+2P_{\langle m,n\rangle})\Gamma_\beta(\beta^{-1}+2P_{\langle -m,-n\rangle})}{\operatorname{Res}\Gamma_\beta(\beta^{-1}+2P_{\langle -m,n\rangle})\Gamma_\beta(\beta^{-1}+2P_{\langle m,-n\rangle})} , \tag{39}$$

where $\operatorname{Res}\Gamma_\beta(x)$ denotes the residue of $\Gamma_\beta$ at a simple pole $x \in -\beta\mathbb{N}-\beta^{-1}\mathbb{N}$. This allows us to write the residues in terms of the functions $d_\pm$ (30)-(31) that enter the four-point structure constants,

$$R_{m,n} = 2P_{\langle m,n\rangle}\frac{\operatorname{Res}d_+(P_{\langle m,-n\rangle})}{d_+(P_{\langle m,n\rangle})} = -2P_{\langle m,n\rangle}\frac{\operatorname{Res}d_-(P_{\langle m,-n\rangle})}{d_-(P_{\langle m,n\rangle})} . \tag{40}$$

(Beware that if $f(x)$ has a pole at $x = ax_0$ and $\tilde{f}(x) = f(ax)$, then $\operatorname{Res}\tilde{f}(x_0) = \frac{1}{a}\operatorname{Res}f(ax_0)$.) It is not completely obvious that these expressions for $R_{m,n}$ have the right signs. In particular, the prefactor $e^{i\pi P_s^2}$ in $d_+(P_s)$ of $d_+(P_s)$ leads to a sign factor $(-1)^{mn}$ in the ratio that appears in eq. (40). But this sign factor is also present in our original definition (36) of $R_{m,n}$, due to the relation

$$\prod_{r\overset{2}{=}1-m}^{m-1}\prod_{s\overset{2}{=}1-n}^{n-1}(P_2-P_1+P_{\langle r,s\rangle}) = (-1)^{mn}\prod_{r\overset{2}{=}1-m}^{m-1}\prod_{s\overset{2}{=}1-n}^{n-1}(P_1-P_2+P_{\langle r,s\rangle}) . \tag{41}$$

Similar relations between conformal blocks' residues, and Liouville theory structure constants, have already appeared in [7] and [8]. These relations will be useful for two reasons:

1. The residue $R_{m,n}$ now depends on its integer indices via the combinations $P_{\langle m,\pm n\rangle}$. Relations of the type $P_{\langle m,n\rangle} = P_{\langle m',n'\rangle}$ that occur at rational central charges will therefore lead to identities between residues. (See the present Section.)

2. In a four-point function (12), both the structure constants and the conformal blocks can be expressed in terms of the same functions $d_\pm$, which will lead to simplifications. (See Section 4.)

## 3.2 Rational limits of generic conformal blocks

Let us discuss the behaviour of conformal blocks at $\beta^2 = \frac{p}{q}$ (with $p, q$ coprime positive integers) for generic momentums $P_1, \dots, P_4$ and $s$-channel dimension $\Delta$. From their definition as sums over states in the Verma module $\mathcal{V}_\Delta$, we know that such blocks exist so long $\Delta$ does not take a degenerate value $\Delta_{\langle r,s\rangle}$. However, Zamolodchikov's recursive representation diverges at $\beta^2 = \frac{p}{q}$. As functions of $\beta^2$, some terms in that representation indeed have poles from two origins:

- The residues $R_{m,n}$ themselves can have poles due to $P_{\langle q,p\rangle} = 0$.

- Factors of the type $\frac{1}{\Delta_{\langle m,-n\rangle}-\Delta_{\langle m',n'\rangle}}$ can diverge.

All the resulting poles have to cancel, leaving a finite expression for the block at $\beta^2 = \frac{p}{q}$. As a function of $\Delta$, this block is still expected to have poles at the degenerate dimensions $\{\Delta_{\langle r,s\rangle}\}$, but these are now multiple poles, as several degenerate dimensions can now coincide.

Let us illustrate the cancellation of two $\beta^2$-poles, and the resulting appearance of a double $\Delta$-pole, in an example. Given a pair of indices $(r,s) \in [1, q-1] \times [1, p-1]$ in the Kac table, the pole of $H_\Delta(\rho)$ at $\Delta = \Delta_{\langle r,s\rangle} = \Delta_{\langle q-r,p-s\rangle}$ receives the following two contributions at the level $N = pq + qs - pr$:

$$H^N_\Delta = \frac{R_{r,s} R_{q-r,p+s}}{(\Delta - \Delta_{\langle r,s\rangle})(\Delta_{\langle r,-s\rangle} - \Delta_{\langle q-r,p+s\rangle})}$$
$$+ \frac{R_{q-r,p-s} R_{2q-r,s}}{(\Delta - \Delta_{\langle q-r,p-s\rangle})(\Delta_{\langle q-r,s-p\rangle} - \Delta_{\langle 2q-r,s\rangle})} + \cdots . \quad (42)$$

Both contributions become infinite at $\beta^2 = \frac{p}{q}$ due to identities of degenerate conformal dimensions eq. (5). For example,

$$\Delta_{\langle r,-s\rangle} - \Delta_{\langle q-r,p+s\rangle} = P^2_{\langle r,-s\rangle} - P^2_{\langle q-r,p+s\rangle} = -P_{\langle q,p\rangle} P_{\langle q-2r,p+2s\rangle} , \quad (43)$$

with $P_{\langle q,p\rangle} \underset{\beta^2=\frac{p}{q}}{=} 0$. But the sum in eq. (42) remains finite, as we will now check. In our calculation, we will neglect terms that are manifestly finite, and only keep the divergent terms in each contributions. Using eq. (40), these divergent terms are

$$H^N_\Delta \underset{\beta^2 \to \frac{p}{q}}{\sim} \frac{2P_{\langle r,s\rangle}}{P_{\langle q,p\rangle} d_+(P_{\langle r,s\rangle})} \left[ -\frac{\operatorname{Res} d_+(P_{\langle r,-s\rangle}) \operatorname{Res} d_+(P_{\langle q-r,-p-s\rangle})}{(\Delta - \Delta_{\langle r,s\rangle}) d_+(P_{\langle q-r,p+s\rangle})} \right.$$
$$\left. + \frac{\operatorname{Res} d_+(P_{\langle q-r,s-p\rangle}) \operatorname{Res} d_+(P_{\langle 2q-r,-s\rangle})}{(\Delta - \Delta_{\langle q-r,p-s\rangle}) d_+(P_{\langle 2q-r,s\rangle})} \right] + \cdots . \quad (44)$$

This expression involves some values and residues of $d_+$ that are finite for generic $\beta$, but become infinite for $\beta^2 \to \frac{p}{q}$. In particular, the residues at $P_{\langle q-r,-p-s\rangle}$ and $P_{\langle 2q-r,-s\rangle}$ both become infinite, because the corresponding poles coincide. Introducing the second-order residue $\operatorname{Res}_2 d_+(P_{\langle 2q-r,-s\rangle}) = \lim_{P \to P_{\langle 2q-r,-s\rangle}} \lim_{\beta^2 \to \frac{p}{q}} (P - P_{\langle 2q-r,-s\rangle})^2 d_+(P)$, we find

$$d_+(P_{\langle q-r,p+s\rangle}) \underset{\beta^2 \to \frac{p}{q}}{\sim} -\frac{\operatorname{Res} d_+(P_{\langle r,-s\rangle})}{P_{\langle q,p\rangle}} , \quad (45)$$

$$d_+(P_{\langle 2q-r,s\rangle}) \underset{\beta^2 \to \frac{p}{q}}{\sim} \frac{\operatorname{Res} d_+(P_{\langle q-r,s-p\rangle})}{P_{\langle q,p\rangle}} , \quad (46)$$

$$\operatorname{Res} d_+(P_{\langle q-r,-p-s\rangle}) \underset{\beta^2 \to \frac{p}{q}}{\sim} -\frac{\operatorname{Res}_2 d_+(P_{\langle 2q-r,-s\rangle})}{P_{\langle q,p\rangle}} , \quad (47)$$

$$\operatorname{Res} d_+(P_{\langle 2q-r,-s\rangle}) \underset{\beta^2 \to \frac{p}{q}}{\sim} \frac{\operatorname{Res}_2 d_+(P_{\langle 2q-r,-s\rangle})}{P_{\langle q,p\rangle}} . \quad (48)$$

This leads to

$$H^N_\Delta \underset{\beta^2 \to \frac{p}{q}}{\sim} \frac{2P_{\langle r,s\rangle} \operatorname{Res}_2 d_+(P_{\langle 2q-r,-s\rangle})}{P_{\langle q,p\rangle} d_+(P_{\langle r,s\rangle})} \left[ -\frac{1}{\Delta - \Delta_{\langle r,s\rangle}} + \frac{1}{\Delta - \Delta_{\langle q-r,p-s\rangle}} \right] + \cdots . \quad (49)$$

Taking the limit, we obtain the manifestly finite expression

$$H_\Delta^N \underset{\beta^2 \to \frac{p}{q}}{\sim} -\frac{4P_{\langle r,s \rangle}^2 \operatorname{Res}_2 d_+(P_{\langle 2q-r,-s \rangle})}{d_+(P_{\langle r,s \rangle})(\Delta - \Delta_{\langle r,s \rangle})^2} + \cdots, \tag{50}$$

which now involves a double pole at $\Delta = \Delta_{\langle r,s \rangle}$. We have therefore determined the residue of this double pole, while neglecting terms that only have a simple pole.

The lowest level double pole of this type occurs for $(p,q) = (3,2)$ and $(r,s) = (1,1)$, at the level $N = 5$. In this case, the two divergent terms are

$$H_\Delta^5 = \frac{R_{1,1}R_{1,4}}{(\Delta - \Delta_{\langle 1,1 \rangle})(\Delta_{\langle 1,-1 \rangle} - \Delta_{\langle 1,4 \rangle})} + \frac{R_{1,2}R_{3,1}}{(\Delta - \Delta_{\langle 1,2 \rangle})(\Delta_{\langle 1,-2 \rangle} - \Delta_{\langle 3,1 \rangle})} + \cdots. \tag{51}$$

Let us introduce the function $\lambda(P) = \prod_\pm (P_2 \pm P_1 + P)(P_3 \pm P_4 + P)$. From the original definition of the residues (36), we have

$$R_{1,1} = -\frac{1}{2}\lambda(0), \tag{52}$$

$$R_{1,2} = \frac{1}{4(\beta^{-4} - 1)}\prod_\pm \lambda\left(\pm\frac{1}{2\beta}\right), \tag{53}$$

$$R_{3,1} = -\frac{1}{24(\beta^4 - 1)(4\beta^4 - 1)}\lambda(0)\prod_\pm \lambda(\pm\beta), \tag{54}$$

$$R_{1,4} = \frac{1}{288(9\beta^{-4} - 1)(4\beta^{-4} - 1)(\beta^{-4} - 1)}\prod_\pm \lambda\left(\pm\frac{1}{2\beta}\right)\lambda\left(\pm\frac{3}{2\beta}\right). \tag{55}$$

Manifestly, $R_{1,1}R_{1,4}$ and $R_{1,2}R_{3,1}$ involve the same $\lambda$-factors if $\beta^2 = \frac{3}{2}$. And we obtain a double pole at $\Delta = 0$,

$$H_\Delta^5 \underset{\beta^2 \to \frac{3}{2}}{\sim} \frac{3}{11200\Delta^2}\lambda(0)\prod_\pm \lambda\left(\pm\frac{1}{2\beta}\right)\lambda(\pm\beta) + \cdots. \tag{56}$$

Another type of pole cancellation occurs between the two terms

$$H_\Delta^{q(p+1)} = \frac{R_{q,p+1}}{\Delta - \Delta_{\langle q,p+1 \rangle}} + \frac{R_{q,p-1}R_{2q,1}}{(\Delta - \Delta_{\langle q,p-1 \rangle})(\Delta_{\langle q,1-p \rangle} - \Delta_{\langle 2q,1 \rangle})} + \cdots, \tag{57}$$

where the residue $R_{q,p+1}$ becomes infinite while the denominator $\Delta_{\langle q,1-p \rangle} - \Delta_{\langle 2q,1 \rangle}$ vanishes. Using the expression (40) for the residues $R_{m,n}$, and neglecting manifestly finite contributions, we have

$$H_\Delta^{q(p+1)} \underset{\beta^2 \to \frac{p}{q}}{\sim} \frac{2P_{\langle 0,1 \rangle}}{d_+(P_{\langle 0,1 \rangle})}\left[\frac{\operatorname{Res} d_+(P_{\langle q,-p-1 \rangle})}{\Delta - \Delta_{\langle q,p+1 \rangle}} \right.$$
$$\left. +\frac{\operatorname{Res} d_+(P_{\langle q,1-p \rangle})\operatorname{Res} d_+(P_{\langle 2q,-1 \rangle})}{P_{\langle q,p \rangle}d_+(P_{\langle 2q,1 \rangle})(\Delta - \Delta_{\langle q,p-1 \rangle})}\right] + \cdots. \tag{58}$$

The value $d_+(P_{\langle 2q,1 \rangle})$, and the residues $\operatorname{Res} d_+(P_{\langle q,-p-1 \rangle})$ and $\operatorname{Res} d_+(P_{\langle 2q,-1 \rangle})$, actually diverge as $\beta^2 \to \frac{p}{q}$, and we find

$$H_\Delta^{q(p+1)} \underset{\beta^2 \to \frac{p}{q}}{\sim} \frac{2P_{\langle 0,1 \rangle}\operatorname{Res}_2 d_+(P_{\langle 2q,-1 \rangle})}{P_{\langle q,p \rangle}d_+(P_{\langle 0,1 \rangle})}\left[-\frac{1}{\Delta - \Delta_{\langle q,p+1 \rangle}} + \frac{1}{\Delta - \Delta_{\langle q,p-1 \rangle}}\right] + \cdots. \tag{59}$$

Taking the limit, we obtain the double pole term,

$$H_\Delta^{q(p+1)} \underset{\beta^2 \to \frac{p}{q}}{\sim} -\frac{8P_{\langle 0,1\rangle}^2 \operatorname{Res}_2 d_+(P_{\langle 2q,-1\rangle})}{d_+(P_{\langle 0,1\rangle})(\Delta - \Delta_{\langle 0,1\rangle})^2} + \cdots . \tag{60}$$

The lowest level double pole of this type occurs for $(p,q) = (2,1)$, at the level 3. In this case, our two divergent terms are

$$H_\Delta^3 = \frac{R_{1,3}}{\Delta - \Delta_{\langle 1,3\rangle}} + \frac{R_{1,1}R_{2,1}}{(\Delta - \Delta_{\langle 1,1\rangle})(\Delta_{\langle 1,-1\rangle} - \Delta_{\langle 2,1\rangle})} + \cdots . \tag{61}$$

The residues that appear in these terms can be deduced from eqs. (52)-(54) via the identity $R_{n,m} = R_{m,n}\big|_{\beta \to \beta^{-1}}$, and we find

$$H_\Delta^3 \underset{\beta^2 \to 2}{\sim} \frac{1}{36\Delta^2}\lambda(0) \prod_\pm \lambda\left(\pm\frac{1}{\beta}\right) + \cdots . \tag{62}$$

We have therefore shown how some terms in the recursive representation diverge in the rational limit, how these divergences cancel when the terms are added, and how this yields conformal blocks with double poles. At higher levels, we expect similar cancellations to yield poles of arbitrary orders. It would be interesting to find a recursive representation of blocks for rational central charges that would be manifestly finite, and that would explicitly exhibit these higher-order poles. This may involve the combinatorial structures that appear when recovering minimal model characters from the recursive representation of torus blocks [9]. It may also be useful to notice that the double pole term of $H_\Delta^{q(p+1)}$ (60) has the same expression (up to a factor 2) as the double pole term of $H_\Delta^{pq+qs-pr}$ (50), if we set $(r,s) = (0,1)$. At rational central charges, the dimensions $\Delta_{\langle r,s\rangle}$ are degenerate for any $(r,s) \in \mathbb{Z}^2$, and it might be simpler to represent conformal blocks as sums over $(r,s) \in \mathbb{Z}^2$, rather than over strictly positive integers.

## 3.3 Degenerate representations

As a function of $\Delta$, the conformal block $\mathcal{F}_\Delta$ has poles for $\Delta = \Delta_{\langle r,s\rangle}$ with $r,s \in \mathbb{N}^*$, as is manifest in Zamolodchikov's recursive representation. Nevertheless, $\mathcal{F}_{\Delta_{\langle r,s\rangle}}$ must be finite and well-defined whenever it appears in a minimal model or generalized minimal model. In such cases, we expect that the dimensions $\Delta_1, \ldots, \Delta_4$ of the four fields are such that the residue of the pole actually vanishes.

This is easy to check in the case of generalized minimal models. For an arbitrary central charge, let us consider a four-point function of four degenerate fields $\left\langle \prod_{i=1}^4 V_{\langle r_i,s_i\rangle} \right\rangle$, and an $s$-channel field $V_{\langle r,s\rangle}$ that obeys the fusion rules (18). Let us compute the residue $R_{r,s}$ of the corresponding pole of the conformal block, using the formula (36). The fusion rules imply $1 - r \le r_1 - r_2 \le 1 + r$ and $1 - s \le s_1 - s_2 \le 1 + s$, so that the residue has a factor $P_2 - P_1 + P_{\langle r_1-r_2,s_1-s_2\rangle} = 0$. Therefore $R_{r,s} = 0$, and the conformal block $\mathcal{F}_{\Delta_{\langle r,s\rangle}}$ is not only finite, but also computable using Zamolodchikov's recursive representation.

Then let us consider a four-point function of four degenerate fields that belong to the Kac table of a minimal model with $\beta^2 = \frac{p}{q}$. As we saw in Section 3.2, the conformal block $\mathcal{F}_\Delta$ in general has higher-order poles that result from the coincidence of several simple poles, and we do not have explicit expressions for the residues. Since the minimal model has finite four-point functions, we nevertheless expect that the residues vanish whenever $\Delta = \Delta_{\langle r,s\rangle}$ obeys the fusion rules, i.e. whenever $(r,s)$ appears in both fusion products $\mathcal{R}_{\langle r_1,s_1\rangle} \times \mathcal{R}_{\langle r_2,s_2\rangle}$ and $\mathcal{R}_{\langle r_3,s_3\rangle} \times \mathcal{R}_{\langle r_4,s_4\rangle}$.

Let us discuss how $\mathcal{F}_\Delta$ behaves as a function of the central charge. This raises the issue of continuing $\mathcal{F}_\Delta$ beyond $\beta^2 = \frac{p}{q}$. We certainly want the four dimensions $\Delta_1, \ldots, \Delta_4$

to remain degenerate, but this does not uniquely determine how they should be continued: the coincidence $\Delta_{\langle r_1,s_1\rangle} = \Delta_{\langle q-r_1,p-s_1\rangle}$ at $\beta^2 = \frac{p}{q}$ leaves us with two different continuations of $\Delta_{\langle r_1,s_1\rangle}$. (We refrain from considering the continuations that are suggested by the coincidences (5).) Taken together, the four dimensions $\Delta_1, \ldots, \Delta_4$ therefore have 16 degenerate continuations. However, for half of these continuations, the fusion products $\mathcal{R}_{\langle r_1,s_1\rangle} \times \mathcal{R}_{\langle r_2,s_2\rangle}$ and $\mathcal{R}_{\langle r_3,s_3\rangle} \times \mathcal{R}_{\langle r_4,s_4\rangle}$ (18) have zero intersection. We eliminate such continuations, by assuming that the indices $(r_i,s_i)$ of our four degenerate fields obey

$$\sum_{i=1}^4 r_i \equiv 0 \bmod 2 \qquad \text{and} \qquad \sum_{i=1}^4 s_i \equiv 0 \bmod 2 . \tag{63}$$

This defines 8 possible continuations of $\mathcal{F}_\Delta$ to arbitrary central charges: let $\mathcal{F}_\Delta^{(\beta^2)}$ be one such continuation. Let $(r,s)$ be a pair of Kac table indices that appears in the fusion products $\mathcal{R}_{\langle r_1,s_1\rangle} \times \mathcal{R}_{\langle r_2,s_2\rangle}$ and $\mathcal{R}_{\langle r_3,s_3\rangle} \times \mathcal{R}_{\langle r_4,s_4\rangle}$ (13) at $\beta^2 = \frac{p}{q}$. Then either $(r,s)$ or $(q-r,p-s)$ appears in the fusion products $\mathcal{R}_{\langle r_1,s_1\rangle} \times \mathcal{R}_{\langle r_2,s_2\rangle}$ and $\mathcal{R}_{\langle r_3,s_3\rangle} \times \mathcal{R}_{\langle r_4,s_4\rangle}$ (18) at generic $\beta^2$: we assume without loss of generality that it is $(r,s)$. Then $\mathcal{F}_{\Delta_{\langle r,s\rangle}}^{(\beta^2)}$ is a conformal block in generalized minimal models, which continues the minimal model conformal block $\mathcal{F}_{\Delta_{\langle r,s\rangle}}^{(\frac{p}{q})}$, and this suggests $\lim_{\beta^2 \to \frac{p}{q}} \mathcal{F}_{\Delta_{\langle r,s\rangle}}^{(\beta^2)} = \mathcal{F}_{\Delta_{\langle r,s\rangle}}^{(\frac{p}{q})}$.

We summarize our expectations as the

**Conjecture 3.1.** *Let* $\mathcal{F}_{\Delta_{\langle r,s\rangle}}^{(\frac{p}{q})}$ *be a minimal model conformal block, and* $\mathcal{F}_\Delta^{(\beta^2)}$ *a continuation to arbitrary central charges and s-channel dimensions, with four degenerate fields that obey eq. (63) and are such that* $\mathcal{R}_{\langle r,s\rangle}$ *is allowed by fusion. Then* $\mathcal{F}_\Delta^{(\beta^2)}$ *is analytic with respect to both* $\Delta$ *and* $\beta^2$ *in the neighbourhood of the finite value* $\mathcal{F}_{\Delta_{\langle r,s\rangle}}^{(\frac{p}{q})}$, *and in particular*

$$\mathcal{F}_{\Delta_{\langle r,s\rangle}}^{(\frac{p}{q})} = \lim_{\Delta \to \Delta_{\langle r,s\rangle}} \mathcal{F}_\Delta^{(\frac{p}{q})} = \lim_{\beta^2 \to \frac{p}{q}} \mathcal{F}_{\Delta_{\langle r,s\rangle}}^{(\beta^2)} . \tag{64}$$

This raises the issues of proving the Conjecture from the definition of conformal blocks, and of finding a generalization of the recursive representation where these equalities manifestly hold.

## 4 Rational limits of non-diagonal four-point functions

Let us investigate how four-point functions of the odd and even CFTs behave when we take the limit $\beta^2 \to \frac{p}{q}$, where $0 < p < q$ are coprime integers.

Zamolodchikov's recursive representation (34) shows that the conformal block $\mathcal{F}_\Delta$ has poles at $\Delta = \Delta_{\langle r,s\rangle}$ for $r,s \in \mathbb{N}^*$. In the odd CFT, non-diagonal fields have dimensions of the type $\Delta = \Delta_{\langle r_1,s_1\rangle}$ with $r_1 \in 2\mathbb{Z}$ and $s_1 \in \mathbb{Z} + \frac{1}{2}$. If $\Delta_{\langle r_1,s_1\rangle} = \Delta_{\langle r,s\rangle}$, then $pr_1 - qs_1 = \pm(pr - qs)$, which implies that $q$ is even. Similarly, a field in the non-diagonal spectrum $\mathcal{S}_{\mathbb{Z}+\frac{1}{2},2\mathbb{Z}}$ of the even CFT can have a diverging conformal block only if $p$ is even. For a number of rational

values of $\beta^2$, let us indicate which CFT (if any) has potential divergences:

| $c$ | $\beta^2$ | Related model | Odd CFT | Even CFT |
|---|---|---|---|---|
| $-\frac{22}{5}$ | $\frac{2}{5}$ | Yang–Lee singularity | finite | ✓ |
| $-2$ | $\frac{1}{2}$ | Spanning tree | ✓ | finite |
| $0$ | $\frac{2}{3}$ | Percolation | finite | ✓ |
| $\frac{1}{2}$ | $\frac{3}{4}$ | Ising model | ✓ | finite |
| $\frac{7}{10}$ | $\frac{4}{5}$ | Tricritical Ising model | finite | ✓ |
| $\frac{4}{5}$ | $\frac{5}{6}$ | Three-state Potts model | ✓ | finite |
| $1$ | $1$ | Four-state Potts model | finite | finite |

(65)

In particular, the odd CFT behaves smoothly as $\beta^2 \to \frac{2}{3}$ i.e. $c \to 0$, and both CFTs are regular as $\beta^2 \to 1$ i.e. $c \to 1$.

A particular four-point function may or may not actually diverge at a potential singularity. Two mechanisms can cancel potential singularities:

- Poles of conformal blocks can have vanishing residues, as happens in minimal models.

- The behaviour of structure constants can make a four-point function finite even when the conformal blocks diverge.

In this Section, we will first investigate the behaviour of structure constants, and then distinguish two cases: a singular case where these two mechanisms do not occur, and a minimal case of four-point functions that have finite limits. These finite limits moreover coincide with four-point functions in minimal models.

### 4.1  Zeros and poles of structure constants

Our four-point structure constants (29) are written in terms of the double Gamma function. For irrational values of $\beta^2$, the function $\Gamma_\beta(x)$ has simple poles for $x \in -\beta\mathbb{N} - \beta^{-1}\mathbb{N}$. For rational values of $\beta^2$, some of these poles coincide. Since $\Gamma_\beta(x)$ depends smoothly on $\beta$, coincidences of simple poles lead to multiple poles. Let us count the multiplicity $S_{r,s}$ of the pole of the four-point structure constant $D_{\langle r,s\rangle}$ at $P = P_{\langle r,s\rangle}$. The factors that produce poles come from the inverse of the two-point function (25), and they are

$$D_{\langle r,s\rangle} = \prod_{\pm} \Gamma_\beta\left(\beta(1 \pm r) \mp \beta^{-1}s\right) \prod_{\pm} \Gamma_\beta\left(\pm\beta r + \beta^{-1}(1 \pm s)\right) \times \cdots. \tag{66}$$

The conditions for poles to occur are the same as for conformal blocks to have potential divergences: $q$ even in the odd CFT (where $r$ is an even integer), and $p$ even in the even CFT (where $s$ is an even integer). Then we find that the multiplicity is

$$S_{r,s} = \max\left(\left\lceil \frac{|r|}{q} - \frac{1}{2}\right\rceil, \left\lceil \frac{|s|}{p} - \frac{1}{2}\right\rceil\right) + \max\left(\left\lfloor \frac{|r|}{q} + \frac{1}{2}\right\rfloor, \left\lfloor \frac{|s|}{p} + \frac{1}{2}\right\rfloor\right). \tag{67}$$

Let us graphically represent this function:

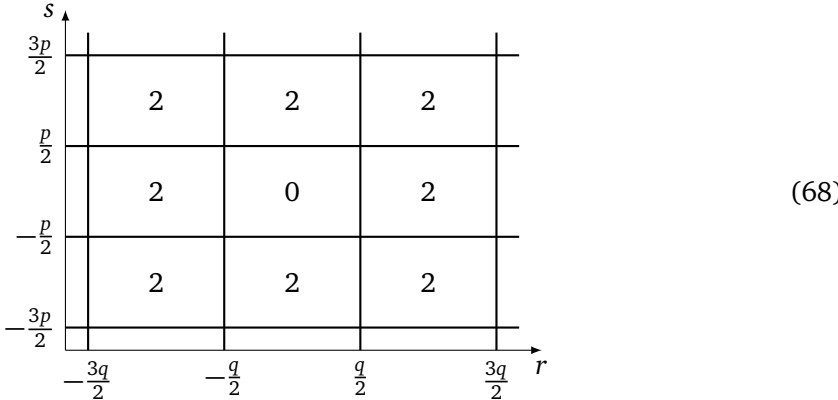

$$(68)$$

The multiplicity vanishes inside the Kac table, and it is $2\max(|m|,|n|)$ in the image of the Kac table under the translation by $(mq, np)$ for $m, n \in \mathbb{Z}$. At a boundary between two images of the Kac table, the multiplicity takes an intermediate value: in particular we have $S_{r,s} = 1$ at the boundary of the Kac table (including at the corners).

Let us discuss the remaining factors of the four-point structure constant (29). These factors can have zeros, but no poles. In the four-point function $\left\langle V_{P_1}^D V_{\langle r_2, s_2 \rangle}^N V_{P_3}^D V_{\langle r_4, s_4 \rangle}^N \right\rangle$ with generic values of $P_1, P_3$, the four-point structure constant $D_{\langle r, s \rangle}$ has no zeros at $\beta^2 = \frac{p}{q}$. We now define a discrete four-point function as a four-point function of the type $\left\langle V_{\langle r_1, s_1 \rangle}^D V_{\langle r_2, s_2 \rangle}^N V_{\langle r_3, s_3 \rangle}^D V_{\langle r_4, s_4 \rangle}^N \right\rangle$, where the indices $r_i, s_i$ obey

$$\begin{cases} r_1 \in \mathbb{Z} \,, \\ r_1 - r_3, r_2, r_4 \in 2\mathbb{Z} \,, \\ s_i \in \mathbb{Z} + \frac{1}{2} \,, \end{cases} \quad \text{or} \quad \begin{cases} r_i \in \mathbb{Z} + \frac{1}{2} \,, \\ s_1 \in \mathbb{Z} \,, \\ s_1 - s_3, s_2, s_4 \in 2\mathbb{Z} \,. \end{cases} \tag{69}$$

A discrete four-point function belongs to the odd or even CFT. In the odd case, $r_1, r_3$ are two integers with the same parity. The four fields belong to the spectrum $\mathcal{S}_{p,q}^{\text{D-series}}$ (8) provided $q > 2\max(|r_i|)$, $p > 2\max(|s_i|)$ and

$$q \in 2r_1 + 2 + 4\mathbb{Z} \text{ (odd case)} \quad \text{or} \quad p \in 2s_1 + 2 + 4\mathbb{Z} \text{ (even case)} \,. \tag{70}$$

Similarly, we define a discrete three-point function as a three-point function that appears in the $s$-channel decomposition of a discrete four-point function, i.e. a three-point function of the type

$$\left\langle V_{\langle r_1, s_1 \rangle}^D V_{\langle r_2, s_2 \rangle}^N V_{\langle r_3, s_3 \rangle}^N \right\rangle \quad \text{with} \quad \begin{cases} r_1 \in \mathbb{Z} \,, \\ r_2, r_3 \in 2\mathbb{Z} \,, \\ s_i \in \mathbb{Z} + \frac{1}{2} \,, \end{cases} \quad \text{or} \quad \begin{cases} r_i \in \mathbb{Z} + \frac{1}{2} \,, \\ s_1 \in \mathbb{Z} \,, \\ s_2, s_3 \in 2\mathbb{Z} \,. \end{cases} \tag{71}$$

(The third field is now the $s$-channel field.) Let us determine whether this vanishes at $\beta^2 = \frac{p}{q}$, assuming that the even integer $p$ or $q$ obeys eq. (70). The formula (26) for the three-point function involves the double Gamma function, and we have

$$\frac{1}{\Gamma_\beta (\beta r - \beta^{-1} s)} \Big|_{\beta^2 = \frac{p}{q}} = 0 \quad \Longleftrightarrow \quad \exists \lambda \in \mathbb{C}, \begin{cases} \lambda q + r \in -\mathbb{N} \,, \\ \lambda p + s \in \mathbb{N} \,. \end{cases} \tag{72}$$

It follows that our three-point function vanishes if and only if

$$\exists \lambda \in 2\mathbb{Z} + 1 \,, \ \exists (\epsilon_1, \epsilon_2, \epsilon_3) \in \{\pm 1\}^3, \begin{cases} \lambda \frac{q}{2} + \epsilon_1 (r_1 + \epsilon_2 r_2 + \epsilon_3 r_3) \in -2\mathbb{N} - 1 \,, \\ \lambda \frac{p}{2} + \epsilon_1 s_1 + \epsilon_2 s_2 + \epsilon_3 s_3 \in 2\mathbb{N} + 1 \,, \end{cases} \tag{73}$$

where the complex number $\lambda$ must actually be an odd integer due to eq. (70). Doing sign reversals, this can equivalently be rewritten as

$$\exists \lambda \in 2\mathbb{Z}+1 \ , \ \exists (\epsilon_1, \epsilon_2, \epsilon_3) \in \{\pm 1\}^3, \ \begin{cases} \lambda \frac{q}{2} + \epsilon_1 (r_1 - \epsilon_2 r_2 - \epsilon_3 r_3) \in 2\mathbb{N}+1 \ , \\ \lambda \frac{p}{2} + \epsilon_1 s_1 + \epsilon_2 s_2 + \epsilon_3 s_3 \in -2\mathbb{N}-1 \ . \end{cases} \tag{74}$$

Let us compare this condition to the fusion rules (14). In the fusion rules, the condition on $r_i$ can equivalently be rewritten as $\frac{q}{2} + \epsilon_1 (r_1 + \epsilon_2 r_2 + \epsilon_3 r_3) \in 2\mathbb{N}+1$. Moreover, our assumptions on the integers or half-integers $r_i, s_i, p, q$ guarantee that the fusion rules are always obeyed up to signs, i.e. there is a unique $\epsilon \in \{\pm 1\}$ such that

$$\forall (\epsilon_1, \epsilon_2, \epsilon_3) \in \{\pm 1\}^3 \ , \ \epsilon_1 \epsilon_2 \epsilon_3 = \epsilon \implies \begin{cases} \frac{q}{2} + \epsilon_1 (r_1 + \epsilon_2 r_2 + \epsilon_3 r_3) & \in 2\mathbb{Z}+1 \ , \\ \frac{p}{2} + \sum_i \epsilon_i s_i & \in 2\mathbb{Z}+1 \ . \end{cases} \tag{75}$$

Obeying fusion rules now means that the elements of $2\mathbb{Z}+1$ are actually positive. If fusion rules are obeyed, then the condition (73) cannot hold for $\lambda = 1$. By a simple sign reversal, it also cannot hold for $\lambda = -1$. And $|\lambda| \geq 3$ is excluded, because the fusion rules imply that our three fields are in the Kac table, so that $|r_i| < \frac{q}{2}$ and $|s_i| < \frac{p}{2}$. Therefore, the three-point function does not vanish.

Conversely, let us assume that fusion rules are violated. For definiteness, assume that the second condition in eq. (14) is violated, so that $\frac{p}{2} + \sum_i \epsilon_i s_i < 0$ for some $\epsilon_i$. Let us also assume that the three-point function is not zero. According to eq. (74) with $\lambda = 1$, we must have $\frac{q}{2} + \epsilon_1 (r_1 - \epsilon_2 r_2 - \epsilon_3 r_3) < 0$. Then according to eq. (73) with again $\lambda = 1$, we must have $\frac{p}{2} + \epsilon_1 s_1 - \epsilon_2 s_2 - \epsilon_3 s_3 < 0$. Combining this with our original assumption, we obtain $\frac{p}{2} + \epsilon_1 s_1 < 0$, which implies that the diagonal field $V_{\langle r_1, s_1 \rangle}^D$ is outside the Kac table. To summarize, we have established the

**Proposition 4.1.** *In rational limits of discrete three-point functions* (71) *with one diagonal and two non-diagonal fields,*

$$\text{fusion rules (14) are obeyed} \quad \Longleftrightarrow \quad \begin{cases} \text{the three-point function is nonzero} \ , \\ \text{the diagonal field is in the Kac table} \ . \end{cases}$$

We refrain from counting the zeros of four-point structure constants. This would be not only complicated, but also not especially illuminating: as we will see, the behaviour of four-point functions depends not only on the behaviour of individual terms in the $s$-channel decomposition, but also on cancellations between different terms.

## 4.2 Singular case

We consider the four-point function $\left\langle V_{P_1}^D V_{\langle r_2, s_2 \rangle}^N V_{P_3}^D V_{\langle r_4, s_4 \rangle}^N \right\rangle$ in the even or odd CFT, and take a limit $\beta^2 \to \frac{p}{q}$ where we have potential divergences. For generic $P_1, P_3$, structure constants have poles (and no zeros), conformal blocks have poles, then how does the four-point functions behave?

Let $Z(\rho)$ be our four-point function, after stripping off the factors $\mathcal{F}^{(0)}(\{z_i\})$ from the conformal blocks (33). The $s$-channel decomposition is then

$$Z(\rho) = \sum_{r,s} Z_{r,s}(\rho) = \sum_{r,s} D_{\langle r,s \rangle} \rho^{\Delta_{\langle r,s \rangle}} \bar{\rho}^{\Delta_{\langle r,-s \rangle}} H_{\Delta_{\langle r,s \rangle}}(\rho) H_{\Delta_{\langle r,-s \rangle}}(\bar{\rho}) \ , \tag{76}$$

where $r, s$ are summed over $2\mathbb{Z} \times (\mathbb{Z} + \frac{1}{2})$ or $(\mathbb{Z} + \frac{1}{2}) \times 2\mathbb{Z}$. Let us consider a term $Z_{r,s}(\rho)$ with $(r,s)$ in the Kac table, i.e. $|r| < \frac{q}{2}$ and $|s| < \frac{p}{2}$: according to eq. (67), $D_{\langle r,s \rangle}$ has a finite limit.

However, $H_{\Delta_{\langle r,s \rangle}}$ has poles, starting with a first-order pole due to the term

$$H_{\Delta_{\langle r,s \rangle}} = 1 + \frac{\rho^{(\frac{q}{2}+r)(\frac{p}{2}+s)} R_{\frac{q}{2}+r,\frac{p}{2}+s}}{\Delta_{\langle r,s \rangle} - \Delta_{\langle \frac{q}{2}+r,\frac{p}{2}+s \rangle}} + \cdots . \tag{77}$$

This term appears in the expansion of $H_{\Delta_{\langle r,s \rangle}}$ because $\frac{q}{2} + r, \frac{p}{2} + s \in \mathbb{N}^*$, and it diverges because $\lim_{\beta^2 \to \frac{p}{q}} \left( \Delta_{\langle r,s \rangle} - \Delta_{\langle \frac{q}{2}+r,\frac{p}{2}+s \rangle} \right) = 0$. This pole corresponds to a null vector whose left-moving conformal dimension is, according to eq. (16),

$$\Delta_{\langle r,s \rangle} + (\tfrac{q}{2}+r)(\tfrac{p}{2}+s) \underset{\beta^2=\frac{p}{q}}{=} \Delta_{\langle \frac{q}{2}+r,-\frac{p}{2}-s \rangle} . \tag{78}$$

Due to an analogous contribution from the right-moving conformal block, $Z_{r,s}(\rho)$ actually has a second-order pole, with the left and right dimensions

$$(\Delta, \bar{\Delta}) \underset{\beta^2=\frac{p}{q}}{=} \left( \Delta_{\langle \frac{q}{2}+r,-\frac{p}{2}-s \rangle}, \Delta_{\langle \frac{q}{2}+r,-\frac{p}{2}+s \rangle} \right) \underset{\beta^2=\frac{p}{q}}{=} \left( \Delta_{\langle q+r,-s \rangle}, \Delta_{\langle q+r,s \rangle} \right) . \tag{79}$$

Therefore, our second-order pole, which corresponds to a null descendent of the primary field $V^N_{\langle r,s \rangle}$, has the same dimensions as the primary field $V^N_{\langle q+r,-s \rangle}$. The structure constant $D_{\langle q+r,-s \rangle}$ for that primary field has a second-order pole according to eq. (67). We will now show that our two second-order poles cancel, leaving us with a first-order pole.

Let us consider the sum $T_1$ of our two terms with second-order poles with the dimensions (79),

$$T_1 = D_{\langle q+r,-s \rangle} \rho^{\Delta_{\langle q+r,-s \rangle}} \bar{\rho}^{\Delta_{\langle q+r,s \rangle}}$$

$$+ D_{\langle r,s \rangle} \rho^{\Delta_{\langle r,s \rangle}} \bar{\rho}^{\Delta_{\langle r,s \rangle}} \frac{\rho^{(\frac{q}{2}+r)(\frac{p}{2}+s)} R_{\frac{q}{2}+r,\frac{p}{2}+s}}{\Delta_{\langle r,s \rangle} - \Delta_{\langle \frac{q}{2}+r,\frac{p}{2}+s \rangle}} \frac{\bar{\rho}^{(\frac{q}{2}+r)(\frac{p}{2}-s)} \bar{R}_{\frac{q}{2}+r,\frac{p}{2}-s}}{\Delta_{\langle r,-s \rangle} - \Delta_{\langle \frac{q}{2}+r,\frac{p}{2}-s \rangle}} . \tag{80}$$

Let us define $\epsilon = P_{\langle q,p \rangle}$, so that $\lim_{\beta^2 \to \frac{p}{q}} \epsilon = 0$. We then have

$$\Delta_{\langle r,s \rangle} - \Delta_{\langle \frac{q}{2}+r,\frac{p}{2}+s \rangle} = -\epsilon \left( P_{\langle r,s \rangle} + \tfrac{\epsilon}{4} \right) . \tag{81}$$

We will moreover rewrite the structure constants and the conformal block residues in terms of the functions $d_\pm(P)$, using eqs. (29) and (40). As an additional simplification, we redefine these functions so as to absorb the dependence on $\rho$,

$$d_\pm(P) \to \rho^{\Delta(P)} d_\pm(P) , \tag{82}$$

where $\Delta(P)$ was given in eq. (2). We have refrained from doing this redefinition earlier, because it would have made the structure constants (29) $\rho$-dependent. This redefinition is therefore conceptually awkward, and we treat it as a computational trick. We then obtain

$$T_1 = d_+(P_{\langle q+r,-s \rangle}) \bar{d}_-(P_{\langle q+r,s \rangle})$$

$$- \frac{4}{\epsilon^2} d_+(P_{\langle r,s \rangle}) \bar{d}_-(P_{\langle r,-s \rangle}) \frac{\operatorname{Res} d_+(P_{\langle \frac{q}{2}+r,-\frac{p}{2}-s \rangle})}{d_+(P_{\langle r,s \rangle} + \frac{\epsilon}{2})} \frac{\operatorname{Res} \bar{d}_-(P_{\langle \frac{q}{2}+r,-\frac{p}{2}+s \rangle})}{\bar{d}_-(P_{\langle r,-s \rangle} + \frac{\epsilon}{2})} \prod_\pm \frac{P_{\langle r,\pm s \rangle} + \frac{\epsilon}{2}}{P_{\langle r,\pm s \rangle} + \frac{\epsilon}{4}} . \tag{83}$$

The function $d_+(P)$ has a simple pole at $P = P_{\langle \frac{q}{2}+r,-\frac{p}{2}-s \rangle}$, let us introduce the function $d_+^1(P)$ such that

$$d_+(P) = \frac{d_+^1(P)}{P - P_{\langle \frac{q}{2}+r,-\frac{p}{2}-s \rangle}} . \tag{84}$$

We then have

$$\operatorname{Res} d_+(P_{\langle \frac{q}{2}+r,-\frac{p}{2}-s\rangle}) = d_+^1(P_{\langle q+r,-s\rangle} - \tfrac{\epsilon}{2}) \quad , \quad d_+(P_{\langle q+r,-s\rangle}) = \frac{2}{\epsilon} d_+^1(P_{\langle q+r,-s\rangle}) . \tag{85}$$

If we moreover introduce the analogous function $\bar{d}_-^1(P) = (P - P_{\langle \frac{q}{2}+r,-\frac{p}{2}+s\rangle})\bar{d}_-(P)$, we can write

$$T_1 = \frac{4}{\epsilon^2} d_+^1(P_{\langle q+r,-s\rangle})\bar{d}_-^1(P_{\langle q+r,s\rangle})$$

$$- \frac{4}{\epsilon^2} d_+^1(P_{\langle q+r,-s\rangle} - \tfrac{\epsilon}{2})\bar{d}_-^1(P_{\langle q+r,s\rangle} - \tfrac{\epsilon}{2}) \frac{d_+(P_{\langle r,s\rangle})}{d_+(P_{\langle r,s\rangle} + \tfrac{\epsilon}{2})} \frac{\bar{d}_-(P_{\langle r,-s\rangle})}{\bar{d}_-(P_{\langle r,-s\rangle} + \tfrac{\epsilon}{2})} \prod_{\pm} \frac{P_{\langle r,\pm s\rangle} + \tfrac{\epsilon}{2}}{P_{\langle r,\pm s\rangle} + \tfrac{\epsilon}{4}} . \tag{86}$$

In this form, it is manifest that the double poles cancel as $\epsilon \to 0$. We are left with a simple pole, whose residue is

$$\lim_{\epsilon \to 0} \epsilon T_1 = d_+^1(P_{\langle q+r,-s\rangle})\bar{d}_-^1(P_{\langle q+r,s\rangle})\left[ 2(\log d_+^1)'(P_{\langle q+r,-s\rangle}) \right.$$

$$\left. + 2(\log \bar{d}_-^1)'(P_{\langle q+r,s\rangle}) + 2(\log d_+)'(P_{\langle r,s\rangle}) + 2(\log \bar{d}_-)'(P_{\langle r,-s\rangle}) - \frac{1}{P_{\langle r,s\rangle}} - \frac{1}{P_{\langle r,-s\rangle}} \right]. \tag{87}$$

In particular, the $\rho$-dependence of $d_{\pm}(P)$ (82) leads to logarithmic terms, whose sum is

$$\lim_{\epsilon \to 0} \epsilon T_1 = 8 d_+^1(P_{\langle q+r,-s\rangle})\bar{d}_-^1(P_{\langle q+r,s\rangle})P_{\langle \frac{q}{2}+r,0\rangle} \log(\rho\bar{\rho}) + \cdots . \tag{88}$$

The reason why there are logarithmic terms, is that we have a cancellation of poles between two terms whose conformal dimensions differ by $O(\epsilon)$.

This cancellation of second-order poles is not an isolated incident: actually, any second-order pole from some term of $Z(\rho)$, is resonant with a pole from another term, i.e. has the same left- and right-moving dimensions. Let us show this by enumerating various terms with second-order poles. We will characterize these terms by three integers $(S, B, \bar{B})$ that indicate the respective orders of poles from the structure constant, left-moving and right-moving conformal blocks, with $S + B + \bar{B} = 2$:

- $(0, 1, 1)$: The case that we already dealt with in detail.

- $(0, 2, 0)$: The left-moving block's second-order pole corresponds to a subsingular vector, with the dimension $\Delta_{\langle \frac{3q}{2}+r,s-\frac{p}{2}\rangle}$ or $\Delta_{\langle r-\frac{q}{2},\frac{3p}{2}+s\rangle}$. In the former case, the pole has the dimensions

$$(\Delta, \bar{\Delta}) = \left( \Delta_{\langle \frac{3q}{2}+r,s-\frac{p}{2}\rangle}, \Delta_{\langle r,-s\rangle} \right) \underset{\beta^2 = \frac{p}{q}}{=} \left( \Delta_{\langle q+r,-p+s\rangle}, \Delta_{\langle q+r,p-s\rangle} \right) . \tag{89}$$

So this pole is resonant with the primary field $V_{\langle q+r,-p+s\rangle}^N$, whose structure constant has a second-order pole.

- $(1, 1, 0)$: The case $S_{r,s} = 1$ in eq. (67) corresponds to the edge of the Kac table, for instance $r = \frac{q}{2}$ and $|s| < \frac{p}{2}$. The left-moving block has a first-order pole with the dimensions

$$(\Delta, \bar{\Delta}) = \left( \Delta_{\langle q,-\frac{p}{2}-s\rangle}, \Delta_{\langle \frac{q}{2},-s\rangle} \right) \underset{\beta^2 = \frac{p}{q}}{=} \left( \Delta_{\langle \frac{q}{2},-p-s\rangle}, \Delta_{\langle \frac{q}{2},p+s\rangle} \right) . \tag{90}$$

So this pole is resonant with the primary field $V_{\langle \frac{q}{2},-p-s\rangle}^N$, whose structure constant has a second-order pole.

- $(2, 0, 0)$: In the four previous cases, we have found that any pole of the type $(0, 1, 1)$, $(0, 2, 0)$ or $(1, 1, 0)$ is resonant with a $(2, 0, 0)$ pole. Now the converse is actually true: any $(2, 0, 0)$ pole is resonant with a pole of the type $(S \leq 1, B, \bar{B})$.

And it can be checked that all these resonances lead to cancellations of second-order poles.

Let us study terms with third-order poles. We focus on terms with the left and right dimensions $(\Delta_{\langle \frac{3q}{2}+r, -\frac{p}{2}+s \rangle}, \Delta_{\langle \frac{q}{2}+r, -\frac{p}{2}+s \rangle})$, with $(r, s)$ in the Kac table. Let us write the five relevant terms, while omitting the dependence on $\rho, \bar{\rho}$. These terms are of the types $(0, 2, 1), (0, 2, 1), (2, 1, 0), (2, 1, 0), (2, 0, 1)$:

$$t_1 = D_{\langle r,s \rangle} \frac{R_{\frac{q}{2}+r, \frac{p}{2}+s}}{\Delta_{\langle r,s \rangle} - \Delta_{\langle \frac{q}{2}+r, \frac{p}{2}+s \rangle}} \frac{R_{\frac{3q}{2}+r, \frac{p}{2}-s}}{\Delta_{\langle \frac{q}{2}+r, -\frac{p}{2}-s \rangle} - \Delta_{\langle \frac{3q}{2}+r, \frac{p}{2}-s \rangle}} \frac{\bar{R}_{\frac{q}{2}+r, \frac{p}{2}-s}}{\Delta_{\langle r,-s \rangle} - \Delta_{\langle \frac{q}{2}+r, \frac{p}{2}-s \rangle}} , \tag{91}$$

$$t_2 = D_{\langle r,s \rangle} \frac{R_{\frac{q}{2}-r, \frac{p}{2}-s}}{\Delta_{\langle r,s \rangle} - \Delta_{\langle \frac{q}{2}-r, \frac{p}{2}-s \rangle}} \frac{R_{\frac{q}{2}+r, \frac{3p}{2}-s}}{\Delta_{\langle \frac{q}{2}-r, -\frac{p}{2}+s \rangle} - \Delta_{\langle \frac{q}{2}+r, \frac{3p}{2}-s \rangle}} \frac{\bar{R}_{\frac{q}{2}+r, \frac{p}{2}-s}}{\Delta_{\langle r,-s \rangle} - \Delta_{\langle \frac{q}{2}+r, \frac{p}{2}-s \rangle}} , \tag{92}$$

$$t_3 = D_{\langle q+r,-s \rangle} \frac{R_{\frac{3q}{2}+r, \frac{p}{2}-s}}{\Delta_{\langle q+r,-s \rangle} - \Delta_{\langle \frac{3q}{2}+r, \frac{p}{2}-s \rangle}} , \tag{93}$$

$$t_4 = D_{\langle r,p-s \rangle} \frac{R_{\frac{q}{2}+r, \frac{3p}{2}-s}}{\Delta_{\langle r,p-s \rangle} - \Delta_{\langle \frac{q}{2}+r, \frac{3p}{2}-s \rangle}} , \tag{94}$$

$$t_5 = D_{\langle q+r,-p+s \rangle} \frac{\bar{R}_{\frac{q}{2}+r, \frac{p}{2}-s}}{\Delta_{\langle q+r,-p+s \rangle} - \Delta_{\langle \frac{q}{2}+r, \frac{p}{2}-s \rangle}} . \tag{95}$$

Let us write these expressions in terms of the functions $d_{\pm}$. From $d_{\pm}$ we build the auxiliary functions $r_0, r_1, r_2, r_3$ such that

$$d_+(P) = \frac{r_0(P)}{(P - P_{\langle \frac{3q}{2}+r, -\frac{p}{2}+s \rangle})(P - P_{\langle \frac{q}{2}+r, -\frac{3p}{2}+s \rangle})} , \tag{96}$$

$$= \frac{r_1(P)}{P - P_{\langle \frac{q}{2}+r, -\frac{p}{2}-s \rangle}} , \tag{97}$$

$$= \frac{r_2(P)}{P - P_{\langle \frac{q}{2}-r, -\frac{p}{2}+s \rangle}} , \tag{98}$$

$$\bar{d}_-(P) = \frac{\bar{r}_3(P)}{P - P_{\langle \frac{q}{2}+r, -\frac{p}{2}+s \rangle}} , \tag{99}$$

and we find

$$t_1 = \frac{4}{\epsilon^3} \bar{r}_3(P_{\langle \frac{q}{2}+r, -\frac{p}{2}+s \rangle}) r_0(P_{\langle q+r,-p+s \rangle} + \frac{\epsilon}{2}) \frac{\bar{d}_-(P_{\langle r,-s \rangle})}{\bar{d}_-(P_{\langle r,-s \rangle} + \frac{\epsilon}{2})} \frac{d_+(P_{\langle r,s \rangle})}{d_+(P_{\langle r,s \rangle} + \frac{\epsilon}{2})}$$
$$\times \frac{r_1(P_{\langle q+r,-s \rangle} - \frac{\epsilon}{2})}{r_1(P_{\langle q+r,-s \rangle} + \frac{\epsilon}{2})} \frac{P_{\langle r,s \rangle} + \frac{\epsilon}{2}}{P_{\langle r,s \rangle} + \frac{\epsilon}{4}} \frac{P_{\langle q+r,-s \rangle} + \frac{\epsilon}{2}}{P_{\langle q+r,-s \rangle}} \frac{P_{\langle r,-s \rangle} + \frac{\epsilon}{2}}{P_{\langle r,-s \rangle} + \frac{\epsilon}{4}} , \tag{100}$$

$$t_2 = \frac{4}{\epsilon^3} \bar{r}_3(P_{\langle \frac{q}{2}+r, -\frac{p}{2}+s \rangle}) r_0(P_{\langle q+r,-p+s \rangle} - \frac{\epsilon}{2}) \frac{\bar{d}_-(P_{\langle r,-s \rangle})}{\bar{d}_-(P_{\langle r,-s \rangle} + \frac{\epsilon}{2})} \frac{d_+(P_{\langle r,s \rangle})}{d_+(P_{\langle r,s \rangle} - \frac{\epsilon}{2})}$$

$$\times \frac{r_2(P_{\langle -r,-p+s \rangle} + \frac{\epsilon}{2})}{r_2(P_{\langle -r,-p+s \rangle} - \frac{\epsilon}{2})} \frac{P_{\langle r,s \rangle} - \frac{\epsilon}{2}}{P_{\langle r,s \rangle} - \frac{\epsilon}{4}} \frac{P_{\langle -r,-p+s \rangle} - \frac{\epsilon}{2}}{P_{\langle -r,-p+s \rangle}} \frac{P_{\langle r,-s \rangle} + \frac{\epsilon}{2}}{P_{\langle r,-s \rangle} + \frac{\epsilon}{4}}, \quad (101)$$

$$t_3 = -\frac{8}{\epsilon^3} \bar{r}_3(P_{\langle \frac{q}{2}+r, -\frac{p}{2}+s \rangle} + \frac{\epsilon}{2}) r_0(P_{\langle q+r,-p+s \rangle} + \frac{\epsilon}{2}) \frac{r_1(P_{\langle q+r,-s \rangle})}{r_1(P_{\langle q+r,-s \rangle} + \frac{\epsilon}{2})} \frac{P_{\langle q+r,-s \rangle} + \frac{\epsilon}{2}}{P_{\langle q+r,-s \rangle} + \frac{\epsilon}{4}}, \quad (102)$$

$$t_4 = -\frac{8}{\epsilon^3} \bar{r}_3(P_{\langle \frac{q}{2}+r, -\frac{p}{2}+s \rangle} - \frac{\epsilon}{2}) r_0(P_{\langle q+r,-p+s \rangle} - \frac{\epsilon}{2}) \frac{r_2(P_{\langle -r,-p+s \rangle})}{r_2(P_{\langle -r,-p+s \rangle} - \frac{\epsilon}{2})} \frac{P_{\langle -r,-p+s \rangle} - \frac{\epsilon}{2}}{P_{\langle -r,-p+s \rangle} - \frac{\epsilon}{4}}, \quad (103)$$

$$t_5 = \frac{8}{\epsilon^3} \bar{r}_3(P_{\langle \frac{q}{2}+r, -\frac{p}{2}+s \rangle}) r_0(P_{\langle q+r,-p+s \rangle}) \frac{\bar{d}_-(P_{\langle r,-s \rangle} + \epsilon)}{\bar{d}_-(P_{\langle r,-s \rangle} + \frac{\epsilon}{2})} \frac{P_{\langle r,-s \rangle} + \frac{\epsilon}{2}}{P_{\langle r,-s \rangle} + \frac{3\epsilon}{4}}. \quad (104)$$

In this form, it is easy to see that $\sum_{i=1}^5 t_i \underset{\epsilon \to 0}{=} O(\frac{1}{\epsilon})$, i.e. the third-order and second-order poles cancel. It is also clear that the needed cancellations do not depend on the properties of the functions $d_\pm$, beyond having poles at degenerate momentums. We are thus led to the

**Conjecture 4.2.** *For any positive fraction $\frac{p}{q}$ and any truncation of $\left\langle V_{P_1}^D V_{\langle r_2,s_2 \rangle}^N V_{P_3}^D V_{\langle r_4,s_4 \rangle}^N \right\rangle$ to a given order in $\rho$, there is a neighbourhood of $\beta^2 = \frac{p}{q}$ where the truncation is meromorphic, with at most a first-order pole at $\beta^2 = \frac{p}{q}$.*

While its truncations are meromorphic, the four-point function itself is not, because at higher order in $\rho$ we have poles that are arbitrarily close to $\beta^2 = \frac{p}{q}$. Rather, the four-point function has an essential singularity on the whole line $\beta^2 \in (0, \infty)$. When $\beta^2 = \frac{p}{q}$ approaches an irrational number, the poles occur at increasingly high conformal dimensions due to $p, q \to \infty$, and their residues tend to zero. So we expect that our four-point function is perfectly well-defined for irrational $\beta^2$, just like the toy function

$$\varphi(\beta^2, \rho) = \sum_{p,q=1}^{\infty} \frac{\rho^{pq}}{\beta^2 - \frac{p}{q}}. \quad (105)$$

Even though the four-point function is not meromorphic, there is a natural definition of its residue at $\beta^2 = \frac{p}{q}$, using the expansion in powers of $\rho$. This residue has a logarithmic dependence (88) on $\rho$. We refrain from conjecturing that this residue is the four-point function of a logarithmic CFT, because it is not clear that the residue obeys crossing symmetry. If it existed, that logarithmic CFT would not contain our non-diagonal primary fields $V_{\langle r,s \rangle}^N$, whose contributions to the four-point function are finite, but only some descendents thereof. The spectrum of that CFT would therefore be quite different from the non-diagonal spectrum of the odd CFT.

### 4.3 Back to minimal models

Let us consider a discrete four-point function in the sense of Section 4.1, in a limit $\beta \to \frac{p}{q}$ where the four fields belong to the spectrum of the D-series minimal model. In other words, we consider a four-point function of the type $\left\langle V_{\langle r_1,s_1 \rangle}^D V_{\langle r_2,s_2 \rangle}^N V_{\langle r_3,s_3 \rangle}^D V_{\langle r_4,s_4 \rangle}^N \right\rangle$, where the values of the indices $(r_i, s_i)$ are as in the spectrum $\mathcal{S}_{p,q}^{\text{D-series}}$ (8).

Let us first show that in the $s$-channel decomposition $Z = \sum_{r,s} Z_{r,s}$ eq. (76), all structure constants $D_{\langle r,s \rangle}$ have finite limits. The behaviour of $D_{\langle r,s \rangle}$ as a function of $r,s$ was discussed in Section 4.1: we have poles whose positions do not depend on $r_i, s_i$, as indicated in the plot (68), and zeros whose positions do depend on $r_i, s_i$. For $(r,s)$ in the Kac table, there are no poles, so $D_{\langle r,s \rangle}$ has a finite limit. Going outside the Kac table, we encounter simple and double poles. However, going ouside the Kac table necessarily violates fusion rules, and we have seen that a fusion-violating three-point structure constant has at least one zero. But $D_{\langle r,s \rangle}$ involves two three-point structure constants, which contribute at least two zeros: enough for cancelling the double poles. Further from the Kac table, we encounter zeros and poles of higher orders, with always at least as many zeros as there are poles. (In fact, far from the Kac table, there are of the order of twice as many zeros as poles.)

Furthermore, if $(r,s)$ is not only in the Kac table, but also allowed by fusion, then not only the structure constant $D_{\langle r,s \rangle}$, but also the corresponding conformal block, have finite limits: these limits are the structure constant and conformal block of the minimal model. On the other hand, if fusion rules are violated, then the conformal block can diverge, and actually the term $Z_{r,s}$ can be nonzero or divergent. But we expect that the sum of fusion-violating terms tends to zero:

**Conjecture 4.3.** *If $(r_i, s_i)$ belong to $\mathcal{S}_{p,q}^{D\text{-series}}$, then $\lim_{\beta^2 \to \frac{p}{q}} \left\langle V^D_{\langle r_1, s_1 \rangle} V^N_{\langle r_2, s_2 \rangle} V^D_{\langle r_3, s_3 \rangle} V^N_{\langle r_4, s_4 \rangle} \right\rangle$ exists, and coincides with the corresponding minimal model four-point function.*

This Conjecture follows from Conjecture 4.2, which limits divergences to simple poles, and Proposition 4.1, which supplies two zeros in fusion-violating terms.

For example, let us consider the four-point function $\left\langle V^D_{\langle 0, \frac{1}{2} \rangle} V^N_{\langle 0, \frac{1}{2} \rangle} V^D_{\langle 0, \frac{1}{2} \rangle} V^N_{\langle 0, \frac{1}{2} \rangle} \right\rangle$, which belongs to the odd CFT. The limit of this four-point function as $\beta^2 \to \frac{p}{q}$ depends on the value of $q$:

- If $q \equiv 2 \bmod 4$, then both fields $V^D_{\langle 0, \frac{1}{2} \rangle}$ and $V^N_{\langle 0, \frac{1}{2} \rangle}$ belong to the spectrum $\mathcal{S}_{p,q}^{D\text{-series}}$ (8), and we recover a minimal model four-point function.

- If $q \equiv 0 \bmod 4$, then $V^D_{\langle 0, \frac{1}{2} \rangle}$ no longer belongs to $\mathcal{S}_{p,q}^{D\text{-series}}$. (That spectrum contains for instance $V^D_{\langle 1, \frac{1}{2} \rangle}$.) The limit is therefore singular.

- If $q$ is odd, nothing happens, the four-point function has a finite limit, and the odd CFT retains its infinite spectrum.

## 5 Rational limits of generalized minimal models

Let us consider a continuation of a diagonal minimal model four-point function in the sense of Section 3.3: a four-point function of diagonal degenerate fields $\left\langle \prod_{i=1}^4 V^D_{\langle r_i, s_i \rangle} \right\rangle$, whose indices belong to the Kac table i.e. $(r_i, s_i) \in [1, q-1] \times [1, p-1]$, and moreover obey eq. (63). This generalized minimal model four-point function exists for any complex central charge, and we will investigate its limit as $\beta^2 \to \frac{p}{q}$. Using the fusion rule (18), the $s$-channel decomposition is

$$\left\langle \prod_{i=1}^4 V^D_{\langle r_i, s_i \rangle} \right\rangle = \sum_{r \overset{2}{=} \max(|r_1 - r_2|, |r_3 - r_4|) + 1}^{\min(r_1 + r_2, r_3 + r_4) - 1} \sum_{s \overset{2}{=} \max(|s_1 - s_2|, |s_3 - s_4|) + 1}^{\min(s_1 + s_2, s_3 + s_4) - 1} D_{\langle r,s \rangle} \mathcal{F}_{\Delta_{\langle r,s \rangle}} \bar{\mathcal{F}}_{\Delta_{\langle r,s \rangle}} , \tag{106}$$

where the four-point structure constant is, according to Section 2.3,

$$
D_{\langle r,s \rangle} = \frac{\prod_{\pm,\pm} \Upsilon_\beta \left( \frac{\beta}{2} + \frac{1}{2\beta} + P_{\langle r,s \rangle} \pm P_1 \pm P_2 \right) \prod_{\pm,\pm} \Upsilon_\beta \left( \frac{\beta}{2} + \frac{1}{2\beta} + P_{\langle r,s \rangle} \pm P_3 \pm P_4 \right)}{\prod_{\pm} \Upsilon_\beta (\beta \pm 2P_{\langle r,s \rangle})}.
\tag{107}
$$

## 5.1 Zeros and poles of structure constants

For fixed integers $r, s$, let us consider $\Upsilon_\beta(\beta + 2P_{\langle r,s \rangle})$ as a function of $\beta$. At $\beta^2 = \frac{p}{q}$, this function has a zero with the multiplicity

$$
M_{r,s} = \left| \left\lfloor \frac{s}{p} \right\rfloor - \left\lfloor \frac{r}{q} \right\rfloor \right|.
\tag{108}
$$

We deduce that the structure constant $D_{\langle r,s \rangle}$ has a pole with the multiplicity

$$
S_{r,s} = M_{r,s} + M_{-r,-s} - M_{\frac{r+r_1+r_2-1}{2}, \frac{s+s_1+s_2-1}{2}} - M_{\frac{r+r_3+r_4-1}{2}, \frac{s+s_3+s_4-1}{2}},
\tag{109}
$$

where negative multiplicities mean zeros rather than poles. One might have expected extra terms with reversed signs for some pairs of indices $(r_i, s_i)$, but these terms actually vanish: for example, $M_{\frac{r+r_1-r_2-1}{2}, \frac{s+s_1-s_2-1}{2}} = 0$ due to $\frac{r+r_1-r_2-1}{2} \leq r_1 - 1 < q$ and $\frac{s+s_1-s_2-1}{2} \leq s_1 - 1 < p$.

Let us plot these pole multiplicities as functions of $r, s$. The values of $r, s$ that appear in the decomposition (106) do not necessarily all belong to the Kac table, but to the first 4 copies of the Kac table, i.e. $r < 2q$ and $s < 2p$. The value of the terms $M_{r,s} + M_{-r,-s}$ only depends on the copy,

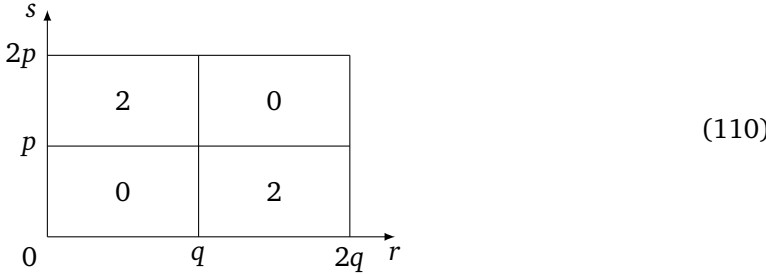

$$
\tag{110}
$$

We do not explicitly show the values at the boundaries of copies of the Kac tables i.e. for $r = q$ or $s = p$: in this case we have $M_{r,s} + M_{-r,-s} = 1$. We then plot the values of the remaining terms $-M_{\frac{r+r_1+r_2-1}{2}, \frac{s+s_1+s_2-1}{2}} - M_{\frac{r+r_3+r_4-1}{2}, \frac{s+s_3+s_4-1}{2}}$:

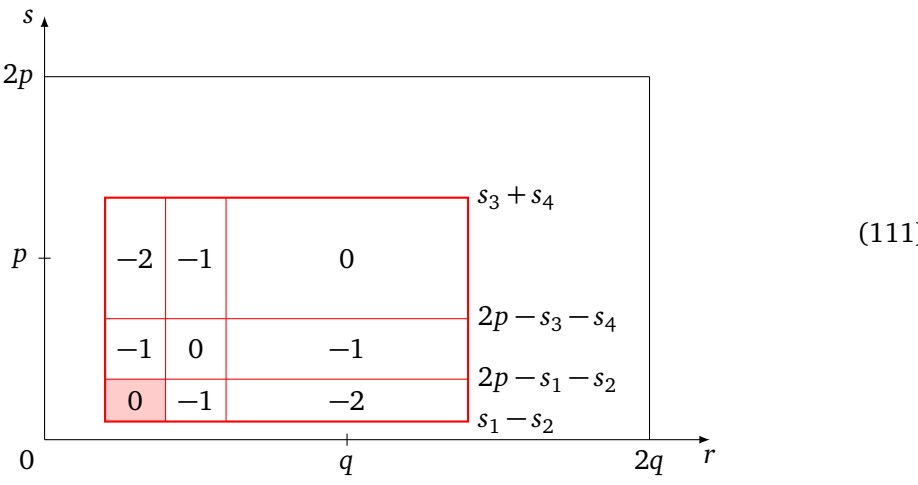

$$
\tag{111}
$$

In this plot the values of $r, s$ on red lines have wrong parities for appearing in the decomposition (106). We indicate the ordinates of the horizontal red lines under assumptions such as $s_1 - s_2 > |s_3 - s_4|$. And we color the region that is allowed by the minimal model's fusion rules (13) in light red. Altogether, the values of the pole multipicities $S_{r,s}$ are as follows:

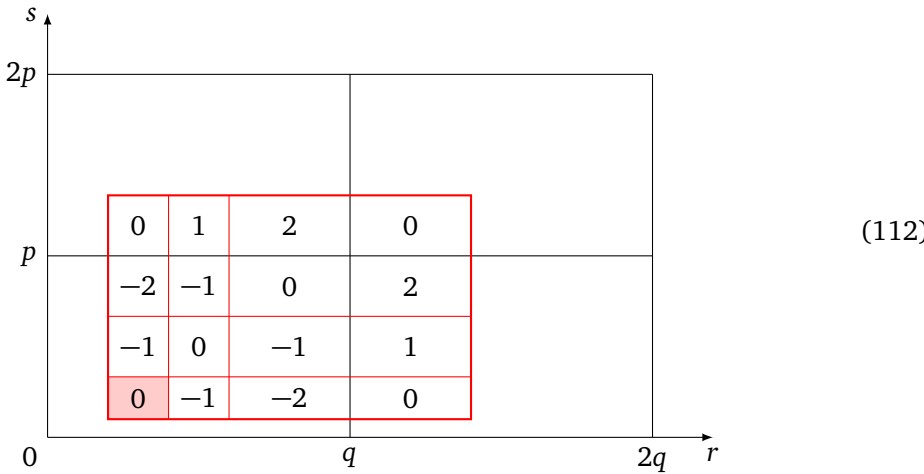

$$(112)$$

Therefore, structure constants have poles if $\min(r_1 + r_2, r_3 + r_4) > q$ and $\min(s_1 + s_2, s_3 + s_4) > p$. Even in the absence of poles, it can happen that $D_{\langle r,s \rangle} \neq 0$ for some $(r, s)$ that violate minimal model fusion rules.

## 5.2 Cancellation of singularities

Thanks to the relation between structure constants and residues of conformal blocks, we expect cancellations between resonant terms, and in particular between resonant terms that have singular limits. Let us illustrate this in the case of the terms whose structure constants have double poles. We consider $r < q$ and $s < p$, such that $D_{\langle r,s \rangle}$ and $D_{\langle 2q-r,2p-s \rangle}$ have finite limits, while $D_{\langle 2q-r,s \rangle}$ and $D_{\langle r,2p-s \rangle}$ have double poles. These four terms are resonant, and they belong to the four top-right regions in Figure 112. Let us compute the combination

$$T_2 = D_{\langle 2q-r,s \rangle} + D_{\langle r,2p-s \rangle}$$
$$+ D_{\langle r,s \rangle} \left( \frac{R_{q-r,p-s}}{\Delta_{\langle r,s \rangle} - \Delta_{\langle q-r,p-s \rangle}} \right)^2 + D_{\langle 2q-r,2p-s \rangle} \left( \frac{R_{q-r,p-s}}{\Delta_{\langle 2q-r,2p-s \rangle} - \Delta_{\langle q-r,p-s \rangle}} \right)^2. \quad (113)$$

Let us write this in terms of functions $d_+, d_-$ using eq. (29) and eq. (40). For this calculation we introduce the notations $\epsilon, P_0, P_1, r_{\pm}(P)$, defined as

$$\epsilon = P_{\langle q,p \rangle} \quad , \quad \begin{cases} P_0 = P_{\langle q-r,p-s \rangle} \\ P_1 = P_{\langle q-r,-p+s \rangle} \end{cases} \quad , \quad d_{\pm}(P) = \frac{r_{\pm}(P)}{P - P_1}. \quad (114)$$

The combination $T_2$ is then rewritten as

$$T_2 = \frac{(r_+ r_-)(P_1 - \epsilon) + (r_+ r_-)(P_1 + \epsilon)}{\epsilon^2}$$
$$- \frac{(r_+ r_-)(P_1)}{\epsilon^2} \left[ \frac{(d_+ d_-)(P_0 - \epsilon)}{(d_+ d_-)(P_0)} \frac{P_0^2}{(P_0 - \frac{\epsilon}{2})^2} + \frac{(d_+ d_-)(P_0 + \epsilon)}{(d_+ d_-)(P_0)} \frac{P_0^2}{(P_0 + \frac{\epsilon}{2})^2} \right]. \quad (115)$$

It is now clear that this has a finite limit as $\epsilon \to 0$: the leading $O(\frac{1}{\epsilon^2})$ terms fall victim to our usual cancellations, while the subleading $O(\frac{1}{\epsilon})$ terms are killed by invariance under $\epsilon \to -\epsilon$.

The finite limit is

$$\lim_{\beta^2 \to \frac{p}{q}} T_2 = (r_+ r_-)''(P_1) + (r_+ r_-)(P_1)\left(\frac{(d_+ d_-)''(P_0)}{(d_+ d_-)(P_0)} - \frac{2}{P_0}\frac{(d_+ d_-)'(P_0)}{(d_+ d_-)(P_0)} + \frac{3}{2P_0^2}\right). \quad (116)$$

This limit is nonzero, and it contains logarithmic terms. More generally, we expect that all singular terms similarly cancel:

**Conjecture 5.1.** *Any continuation of a four-point function of the $(p, q)$ diagonal minimal model, has a finite limit when $\beta^2 \to \frac{p}{q}$.*

Next we will discuss whether this finite limit coincides with the corresponding minimal model four-point function. From the presence of logarithmic terms, we already know that this is not always the case.

## 5.3 Limits of four-point functions

In the decomposition (106) of our four-point function, let us consider a term that is allowed by the minimal model's fusion rules. According to Section 3.3, the corresponding conformal block then tends to a minimal model conformal block. Moreover, the corresponding structure constant also has a finite limit, which coincides with the minimal model structure constant. Therefore, the sum of such terms tends to the minimal model four-point function. The question is whether the sum of the rest of the terms tends to zero. Several situations may occur:

- The limit of our four-point function may disagree with the minimal model. This must happen whenever this limit has logarithmic terms, as in eq. (116). This must also happen whenever there is a term that is disallowed by minimal model fusion rules, that does not resonate with another term, and whose structure constant has a finite limit. For example, let us consider $\left\langle V_{\langle 3,2\rangle}^D V_{\langle 3,2\rangle}^D V_{\langle 2,2\rangle}^D V_{\langle 2,2\rangle}^D \right\rangle$ in the limit $\beta^2 \to \frac{4}{3}$. The $s$-channel fields, and the numbers of poles of their structure constants, are:

| field | $V_{\langle 1,1\rangle}^D$ | $V_{\langle 1,3\rangle}^D$ | $V_{\langle 3,1\rangle}^D$ | $V_{\langle 3,3\rangle}^D$ | |
|---|---|---|---|---|---|
| # poles | 0 | $-1$ | $-1$ | 0 | (117) |

  where the minimal model fusion rules only allow the field $V_{\langle 1,1\rangle}^D$. The structure constants and contributions of $V_{\langle 1,3\rangle}^D$ and $V_{\langle 3,1\rangle}^D$ tend to zero, but the contribution of $V_{\langle 3,3\rangle}^D$ has a finite limit.

- The limit of our four-point function may agree with the minimal model thanks to cancellations between different terms. For example, consider $\left\langle V_{\langle 4,1\rangle}^D V_{\langle 4,1\rangle}^D V_{\langle 4,1\rangle}^D V_{\langle 4,1\rangle}^D \right\rangle$ in the limit $\beta^2 \to \frac{5}{6}$. The $s$-channel fields, and the numbers of poles of their structure constants, are:

| field | $V_{\langle 1,1\rangle}^D$ | $V_{\langle 3,1\rangle}^D$ | $V_{\langle 5,1\rangle}^D$ | $V_{\langle 7,1\rangle}^D$ | |
|---|---|---|---|---|---|
| # poles | 0 | $-2$ | $-1$ | 0 | (118) |

  where the minimal model fusion rules only allow the field $V_{\langle 1,1\rangle}^D$. The contributions of $V_{\langle 3,1\rangle}^D$ and $V_{\langle 7,1\rangle}^D$ both have finite limits, but they are resonant and actually cancel.

- The limit of our four-point function may agree with the minimal model because our fusion rules coincide with minimal model fusion rules. For a given four-point function, this happens whenever $p, q$ are large enough, thanks to our assertion in Section 2.2 on the limit of minimal model fusion rules for $p, q \to \infty$.

To conclude, let us imagine that we follow a given four-point function of degenerate fields as the central charge varies in the half-line $(-\infty, 1)$. At most rational values of $\beta^2 = \frac{p}{q}$, the integers $p, q$ will be large enough for our four-point function to coincide with the corresponding minimal model four-point function. There may however be rational values of $\beta^2$ where our four-point function differs from the minimal model four-point function. And there will be an infinite but not dense set of rational values of $\beta^2$, such that one or more of our four fields is outside the Kac table. We leave the exploration of this last case for future work.

These results provide more justification for the name *generalized minimal models*. We use this name for diagonal CFTs that exist at any central charge, and whose spectrums are made of all degenerate fields. We now know that these CFTs not only generalize minimal models, but also interpolate between them.

# 6 Acknowledgments

I am grateful to Jesper Jacobsen, Nina Javerzat, Santiago Migliaccio, Marco Picco, Hubert Saleur, and Raoul Santachiara, for stimulating discussions and collaborations on related topics. Moreover, I wish to thank Santiago Migliaccio for many helpful comments on this text, and the anonymous SciPost referee for valuable suggestions.

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
