# Peer review of "On 2d CFTs that interpolate between minimal models"

_SciPost Physics, doi:SciPost Phys. 6, 075 (2019)_

## Round 2 · Referee Report · Anonymous (Referee 1) · 2019-4-30

Strengths

A new approach to finding the most general extension of minimal CFTs at rational values of the central charge, by considering limits of the generic central charges.

Weaknesses

The logic of the paper is slightly complicated and difficult to follow, and the conclusions not easy to draw.
It would be interesting to study a particular case in detail and be able to make more definite statements.

Report

The subject of the paper is the study of two dimensional conformal field theories at arbitrary values of central charge and at the limiting value when the central charges become rational, the purpose being to understand how the minimal models, or generalised minimal models, can be obtained from generic conformal field theories. The study is based on the analytic structure of the four point functions.
The parameter of interest is the dimension of the fields in the intermediate channel; according to a representation due to Zamolodchikov, the four point function as a function of this variable has poles at all the values corresponding to the degenerate fields.
It is the position and the multiplicity of these poles that is the focus of the present work, namely how do they combine in the limit of rational central charge.

The first observation is that for generic charges of the four operators, the Zamolodchikov’s
representation superficially diverges in the rational limit; however the superficial divergences cancel between different terms. The mechanism is illustrated in a simple case, where the simple poles become double poles after the cancelation of divergences - but no general proof is given.
The remaining statements are a couple of conjectures concerning the behaviour of the conformal blocks when the charges of the four external fields are degenerate and the internal dimension approaches the value of a degenerate field allowed or not by the fusion rules.

The results are not extremely conclusive, but the paper is presumably the beginning a systematic study of the rational limit of generic CFTs.

Requested changes

Concerning te structure of the paper 1. it would be useful to summarise the results in the introduction, and try to keep the paper self-contained (e.g. define a non-diagonal CFT). 2. it would be helpful to explain if the summation indexing (2.28) is continuous or discrete, and what happens to this representation for the minimal models. I would suggest also: 3. to use a figure caption for the figure on page 8, 4. to use the standard plural for the late words (e.g momentums -> momenta) 5. to use the standard form for the references (i.e. with the journal number).

---

## Round 3 · Referee Report · Anonymous · 2019-6-6

Report

The author has complied with the suggested changes, so I recommend the paper for publication.

---

## Round 3 · Author Response

I am grateful to the reviewer for valuable suggestions. The reviewer's main suggestion was to make the article clearer and more self-contained, in particular by summarizing the results in the introduction.

I have therefore restructured the article: the new Introduction (Section 1) now includes material from the former Introduction, the former Summary and discussion (Section 6), and whatever in Section 2 was needed for understanding the results. I have also added a subsection 1.3 with a summary of results, which hopefully also clarifies the structure of the article. Section 6 has now disappeared, most of the material moved to Section 1, the specific example of a four-point function however moved to the end of Section 4.

Notice that the formula (1.8) is a new way of writing the spectrums of D-series minimal models,
without distinguishing two cases depending on the parity of $p$. I hope that this makes limits easier to figure out.

---

## Round 3 · List of Changes

Here I am answering the reviewer's numbered suggestions:

1. See my comments above.

2. The formula in question is now called (1.12) instead of (2.28). I have added a couple of sentences on the summation indexing.

3. The figure now has a caption, it is now on page 10 instead of page 8.

4. The plural momentums is less common than momenta but it is correct. For reasons to use regular plurals when possible, see my blog post: http://researchpracticesandtools.blogspot.com/2018/01/will-no-one-rid-me-of-these-tiresome.html

5. The formatting of references is a matter for the journal. Let me however point out that I take the articles I cite from arXiv. I cannot in principle vouch for the correctness and relevance of journal versions if they exist.

---

## Editorial Decision

published